# Single-cell RNA-seq analysis reveals penaeid shrimp hemocyte subpopulations and cell differentiation process

**Keiichiro Koiwai[1,2]\*, Takashi Koyama[3,4], Soichiro Tsuda[5], Atsushi Toyoda[6], Kiyoshi Kikuchi[3], Hiroaki Suzuki[7], Ryuji Kawano[1]**

[1]Department of Biotechnology and Life Science, Tokyo University of Agriculture and Technology, Koganei, Japan; [2]Laboratory of Genome Science, Tokyo University of Marine Science and Technology, Minato, Japan; [3]Fisheries Laboratory, Graduate School of Agricultural and Life Sciences, The University of Tokyo, Hamamatsu, Japan; [4]Graduate School of Fisheries and Environmental Sciences, Nagasaki University, Nagasaki, Japan; [5]bitBiome Inc, Shinjuku, Japan; [6]Advanced Genomics Center, National Institute of Genetics, Mishima, Japan; [7]Department of Precision Mechanics, Faculty of Science and Engineering, Chuo University, Bunkyo, Japan

**Abstract** Crustacean aquaculture is expected to be a major source of fishery commodities in the near future. Hemocytes are key players of the immune system in shrimps; however, their classification, maturation, and differentiation are still under debate. To date, only discrete and inconsistent information on the classification of shrimp hemocytes has been reported, showing that the morphological characteristics are not sufficient to resolve their actual roles. Our present study using single-cell RNA sequencing revealed six types of hemocytes of *Marsupenaeus japonicus* based on their transcriptional profiles. We identified markers of each subpopulation and predicted the differentiation pathways involved in their maturation. We also predicted cell growth factors that might play crucial roles in hemocyte differentiation. Different immune roles among these subpopulations were suggested from the analysis of differentially expressed immune-related genes. These results provide a unified classification of shrimp hemocytes, which improves the understanding of its immune system.

**\*For correspondence:** koiwai@kaiyodai.ac.jp

## Introduction

Aquaculture is an important source of animal protein and is considered one of the most important long-term growth areas of food production, providing 60% of fish for human consumption (*FAO, 2020*) (http://www.fao.org/fishery/statistics/en). However, crustaceans that lack an adaptive immune system (*Jiravanichpaisal et al., 2006*; *Tassanakajon et al., 2013*; *Flegel, 2019*) are vulnerable to pathogens. This means that ordinal vaccination is not applicable to crustaceans, unlike in fish aquaculture. Shrimp is the main target species for crustacean aquaculture. Therefore, an immune priming system for shrimp, which is entirely different from conventional vaccines, needs to be developed to control the infection of pathogens. However, little is known about the immune system of crustaceans due to the lack of biotechnological tools, such as uniform antibodies and other biomarkers (*Zhang et al., 2019*).

Hemocytes, which are immune cells of crustaceans, are traditionally divided into three morphological types based on the dyeing of intracellular granules, which was established by Bauchau and colleagues (*Bauchau, 1981*; *Söderhäll and Smith, 1983*; *Johansson et al., 2000*). However, there have been additional reports on the classification of the hemocytes of shrimp; they were classified into four, eight, and five types based on electron microscopy (*van de Braak et al., 2002*), another

**Figure 1.** The schematic of single-cell mRNA sequencing (scRNA-seq) analysis of penaeid shrimp *M. japonicus* hemocytes. Single hemocytes were analyzed through the microfluidics-based Drop-seq, mRNA sequencing for the preparation of de novo assembled gene sets, in silico analysis workflow, and morphology-based cell classification.

dyeing method (*Kondo et al., 2012*), and iodixanol density gradient centrifugation (*Dantas-Lima et al., 2013*), respectively. As the morphology and dye staining properties of shrimp hemocytes are not absolute indicators, no unified understanding of their role has been established yet. Molecular markers, such as specific mRNAs, antibodies, or lectins, are usually available for characterizing the subpopulations of cells in model organisms, but this is not often the case for non-model organisms. Although monoclonal antibodies have been developed for some hemocytes of shrimp (*Rodriguez et al., 1995*; *van de Braak et al., 2000*; *Sung et al., 1999*; *Sung and Sun, 2002*; *Winotaphan et al., 2005*; *Lin et al., 2007*; *Xing et al., 2017*), their number is lower than that of humans, and their correspondence to the cell type, as well as their differentiation stage, is under debate.

Recently, single-cell mRNA sequencing (scRNA-seq) techniques have dramatically changed this scene, allowing researchers to annotate non-classified cells solely based on the mRNA expression patterns of each cell. In particular, droplet-based mRNA sequencing, such as Drop-seq, developed by Macosko (*Macosko et al., 2015*), has gained popularity for classifying cells and identifying new cell types. The enormous amount of biological data obtained from scRNA-seq leads us to classify cells into specific groups, analyze their heterogeneity, predict the functions of single-cell populations based on the gene expression profiles, and determine the cell proliferation or development pathways based on the pseudo-time ordering of a single cell (*Trapnell et al., 2014*; *Soneson and Robinson, 2018*). More recently, hemocytes of invertebrate, fly, and mosquito models have been subjected to these types of microfluidic-based scRNA-seq to reveal their functions (*Raddi et al., 2020*; *Tattikota et al., 2020*; *Cho et al., 2020*; *Cattenoz et al., 2020*; *Fu et al., 2020*).

Here, we performed scRNA-seq analysis on *Marsupenaeus japonicus* hemocytes to classify the hemocyte types and characterize their functions using the custom-built Drop-seq platform. To perform scRNA-seq, a high-quality gene reference is essential; however, such reference genomes are scarce for crustaceans because of the extremely high proportion of simple sequence repeats (*Zhang et al., 2019*). We circumvented this problem by preparing reference genes using hybrid de novo assembly of short- and long-read RNA sequencing results. The sequences obtained from the scRNA-seq were mapped onto the reference genes successfully. Our scRNA-seq uncovered the transcriptional profiles of a few thousand *M. japonicus* hemocytes. We identified the markers of each population and the differentiation pathways associated with their maturation. We also discovered the cell growth factors that might play crucial roles in hemocyte differentiation. Different immune

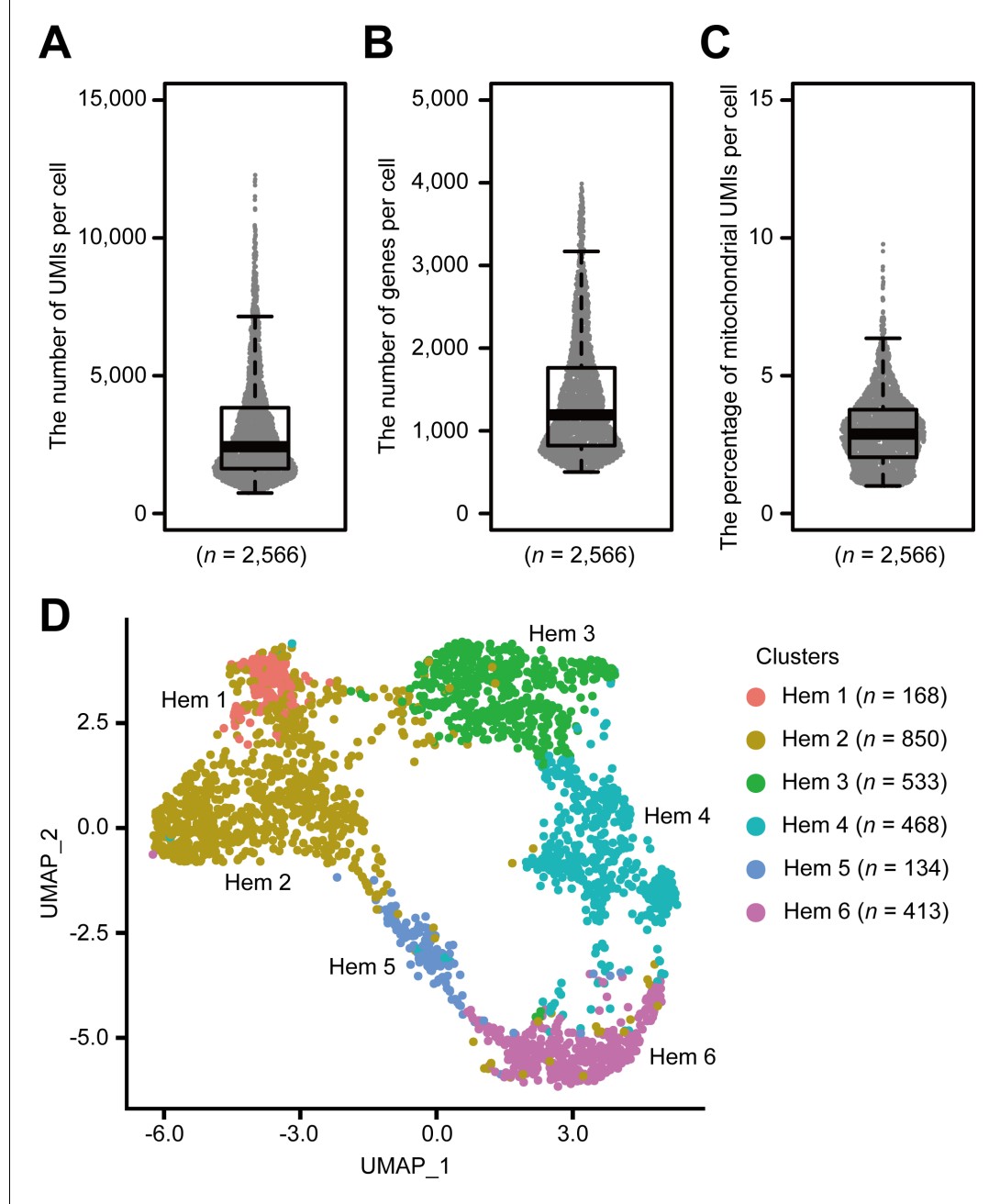

**Figure 2.** Single-cell mRNA sequencing (scRNA-seq) analysis of penaeid shrimp *M. japonicus* hemocytes. Distribution and the median of the number of transcripts (unique molecular identifiers [UMIs]) (**A**), genes (**B**), and percentage of mitochondrial UMIs (**C**) detected per cell. Uniform manifold approximation and projection (UMAP) plot of SCTransform batch corrected and integrated of hemocytes from three shrimps (*n* = 2566) (**D**). scRNA-seq analysis of penaeid shrimp *M. japonicus* hemocytes.

The online version of this article includes the following source data and figure supplement(s) for figure 2:

**Source data 1.** Excel sheets pertaining to UMIs, genes , mitochondrial UMIs detected per hemocyte used for *Figure 2A-C*.

**Figure supplement 1.** Sinaplots show the distribution of the number of transcripts (scored by unique molecular identifiers [UMIs]) (**A**), genes (**B**), and percentage of mitochondrial UMIs (**C**) detected per hemocyte on individual shrimp.

**Figure supplement 1—source data 1.** Excel sheets pertaining to UMIs, genes , mitochondrial UMIs detected per hemocyte on individual shrimp used for *Figure 2—figure supplement 1A-C*.

roles among these subpopulations were also suggested from the analysis of differentially expressed

immune-related genes. Our results present a unified classification of shrimp hemocytes and a deeper understanding of the immune system of shrimp.

## Results

### scRNA-seq clustering of *M. japonicus* hemocytes

Our study utilized scRNA-seq to determine the cellular subtypes with a distinct transcriptional expression (*Figure 1A*). To map the scRNA-seq sequences from *M. japonicus* hemocytes, we first prepared a high-quality de novo assembled reference genes using hybrid assembly of short- and long-read RNA sequencing results. The quality of assembled genes was quality-checked by BUSCO against arthropoda database. BUSCO tool showed 91.5% (35.6% complete and single-copy genes and 55.9% complete and duplicated genes) as complete genes, showing good completeness of the assembly. Some of the assembled genes constructed in this study showed homology to the same genes despite processed EvidentialGene program to remove similar sequences. Cluster database at high identity with tolerance (CD-HIT) programs was also applied on our draft assembled genes, but it was not able to cluster better than EvidentialGene program. Since we thought that subjective clustering of specific genes by ourselves would result in bias, we conducted our analysis based on the results of clustering by EvidentialGene program. Then, by using self-built Drop-seq microfluidic chips, single hemocytes were captured and their mRNA was barcoded using the droplet-based strategy. This process was performed in triplicates for three shrimp individuals. Following library preparation and sequencing, the transcriptomes obtained from scRNA-seq were mapped against the reference genes to discover the cell types.

Using the Drop-seq procedure, we profiled a total of 2566 cells and obtained a median value of 2427 unique molecular identifiers (UMIs), 1193 genes, and 2.9% of mitochondrial genes per cell across three replicates (*Figure 2*, *Figure 2—figure supplement 1A–C*). There was some transcriptional variability, which may have resulted from the artifacts of the Drop-seq system, because it is consistent with the original Drop-seq paper (*Macosko et al., 2015*) and with recent findings in other organisms, such as fish (*Carmona et al., 2017*), flies (*Tattikota et al., 2020*; *Cho et al., 2020*), and mosquitoes (*Raddi et al., 2020*). We constructed three Drop-seq libraries, which accumulates 1500 cells each from three individual shrimps. The sequence depth of read number per single-cell counts about 800,000 reads per cell based on the calculation that 400 million reads from NextSeq 500/550 High Output v2 kit/three individual shrimp/1500 cells. Conventionally, this read number is sufficient for any biological clustering with reference to the additional information of Drop-seq original paper; however, we obtained a median value of 2427 UMIs per cell, which is lower than other organisms such as *Drosophila melanogaster* counting 8000 UMIs (*Tattikota et al., 2020*) or HEK (human) and 3T3 (mouse) cells counting 9000 UMIs (*Macosko et al., 2015*), respectively. This is due to the lack of genomic information of *M. japonicus*; hence, we prepared de novo assembled transcripts for the read mapping. As the genomic analysis of *M. japonicus* further progresses, we believe that these UMI values will be improved.

Applying the SCTransform batch correction method integrated into the Seurat package allowed us to remove the individual differences. SCTransform successfully integrated all three shrimp datasets (*Figure 2—figure supplement 1D*), among which we identified a total of six clusters (*Figure 2D*). Each cluster contained the following number of cells: Hem 1, 168 cells (6.5%); Hem 2, 850 cells (33.1%); Hem 3, 533 cells (20.8%); Hem 4, 468 cells (18.2%); Hem 5, 134 cells (5.2%); and Hem 6, 413 cells (16.1%).

Since not all the de novo assembled transcripts were expressed in this single-cell data, the highly expressed transcript that represents 50% of the total normalized expression data was extracted as 'Ex50' for further analysis. The functions of 1362 transcripts composed Ex50 were predicted by blastx program on penaeid shrimp proteins and by eggNOG-mapper annotation for the eukaryotic orthologous groups (KOGs) and gene ontologies (GOs) (*Supplementary file 1*).

### KOGs and GOs analysis on clusters

Among 1362 transcripts composed Ex50, 865 (70.9%) transcripts on shrimp proteins, 886 (65.1%) transcripts on KOG annotation, and 614 (45.1%) transcripts on GO annotation were annotated.

In the result of KOGs, seven KOG annotations (i.e., chromatin structure and dynamics, amino acid transport and metabolism, lipid transport and metabolism, transcription, replication, recombination and repair, signal transduction mechanisms, and intracellular trafficking, secretion, and vesicular transport) were highly expressed among most of the cells in cluster Hem 1 (*Figure 3A*). In the result of GOs, two GO annotations (i.e., molecular transducer activity and growth) were highly expressed in cluster Hem 1. Furthermore, one GO annotation, immune system process, was highly expressed in cluster Hem 6 (*Figure 3B*). This result showed that a group of genes involved in cell division and metabolism are highly expressed in Hem 1. It was also predicted that immune-related genes were strongly expressed in Hem 6. Genes involved in basic metabolism, such as lipid metabolism and transcription, were expressed in almost all clusters, but among those the expression tended to be stronger in Hem 1.

## Cluster-specific markers and their functional prediction

A total of 415 cluster-specific markers were predicted using the Seurat FindMarkers tool (*Figure 4A*; *Supplementary file 2*). For each cluster, 167 (Hem 1), 29 (Hem 2), 38 (Hem 3), 37 (Hem 4), 42 (Hem 5), and 102 (Hem 6) markers were identified. Important markers in each cluster are described in the dot plot figure (*Figure 4B*).

Many of Hem 1-specific markers were related to cell proliferation, cell migration, and colony formation. For example, *sin3 histone deacetylase corepressor complex component SDS3* (*SUDS3*) (Mj-9135) and *histone deacetylase* (*HDAC*) (Mj-18057) are known to be essential for the proliferation through controlling cell cycle progression, DNA replication and repair, and cell death in mammal studies (*Dannenberg et al., 2005*; *McDonel et al., 2012*). *Polypyrimidine tract-binding protein 1* (*PTBP1*) (Mj-18644) is also a gene related to cell proliferation, cell migration, and colony formation in human tumor studies (*Cheung et al., 2009*; *Shibayama et al., 2009*; *Monzón-Casanova et al., 2020*). It is a multifunctional RNA-binding protein that is overexpressed in glioma, a type of tumor that occurs in the brain, and a decreased expression of PTBP inhibits cell migration and increases the adhesion of cells to fibronectin and vitronectin (*Cheung et al., 2009*; *Shibayama et al., 2009*). PTBP has been shown to be involved in germ cell differentiation in *D. melanogaster* and is essential for the development of *Xenopus laevis* (*Jiao et al., 2016*). *Cell division cycle 7-related protein kinase* (*CDC7*) (Mj-22708 and Mj-22709) is required for continuous DNA replication in mammalian cells, and reduced levels of Cdc7 kinase are viable but exhibit significantly reduced body size and impaired cell proliferation (*Kim et al., 2003*). The high expression of homologues of cell proliferation-related genes suggests that Hem 1 is a cluster related to cell proliferation in shrimp hemocytes.

Among the Hem 2 markers, three markers were annotated with *hemocyte transglutaminase* (*HemTGase*) (Mj-16280, Mj-16290, and Mj-21219), two markers were annotated with *phenoloxidase-activating factor 2-like* (*PPAF2-like*) (Mj-15851 and Mj-23817), and two markers were annotated with *single VWC domain protein 5* (*Vago 5*) (Mj-24191 and Mj-24193). *TGase* is an immature hemocyte marker of crayfish and shrimp. When the extracellular TGase is digested, hemocytes start to differentiate into mature hemocytes (*Lin et al., 2008*; *Huang et al., 2004*; *Lin et al., 2011*). The high expression of *HemTGase* suggests that Hem 2 is in the early stage of hemocytes. PPAF2 has a role to activate phenoloxidase (PO) cascade involving melanization (*Boonchuen et al., 2021*). Vago 5 is a kind of shrimp cytokine functionally similar to interferon and found to be involved in shrimp antiviral immunity (*Li et al., 2015*; *Gao et al., 2019*). PPAF2 and Vago 5 do not directly exhibit defense mechanisms, but work by activating other genes and other immune system. Therefore, Hem 2 may work as an immune activator against other hemocytes.

In cluster Hem 3, two markers were annotated with *anti-lipopolysaccharide factor-A1* (*ALF-A1*) (Mj-19322 and Mj-23779), and one marker was annotated with *viral responsive protein* (*VRP*) (Mj-4390). In cluster Hem 4, one marker was annotated with *anti-lipopolysaccharide factor-like* (*ALF-like*) (Mj-7049), one marker was annotated with *myeloid differentiation factor 2–related lipid-recognition protein* (*ML protein*) (Mj-10040), and three markers were annotated with *penaeidin-II* (Mj-18245, Mj-19281, and Mj-20968). ALF and penaeidin-II are well-studied antimicrobial peptides (AMPs) (*Destoumieux et al., 1997*; *Bachère et al., 2004*; *An et al., 2016*; *Liu et al., 2006*; *Mekata et al., 2010*; *Rosa and Barracco, 2010*; *Rosa et al., 2013*). These AMPs are stored in the granules of hemocytes. VRP protein is distributed in granular-containing hemocytes and induced its expression by virus infection (*Elbahnaswy et al., 2017*). ML protein recognizes a lipid component of virus and induces the expression of Vago 5 (*Gao et al., 2019*), which is highly expressed in Hem 2. The

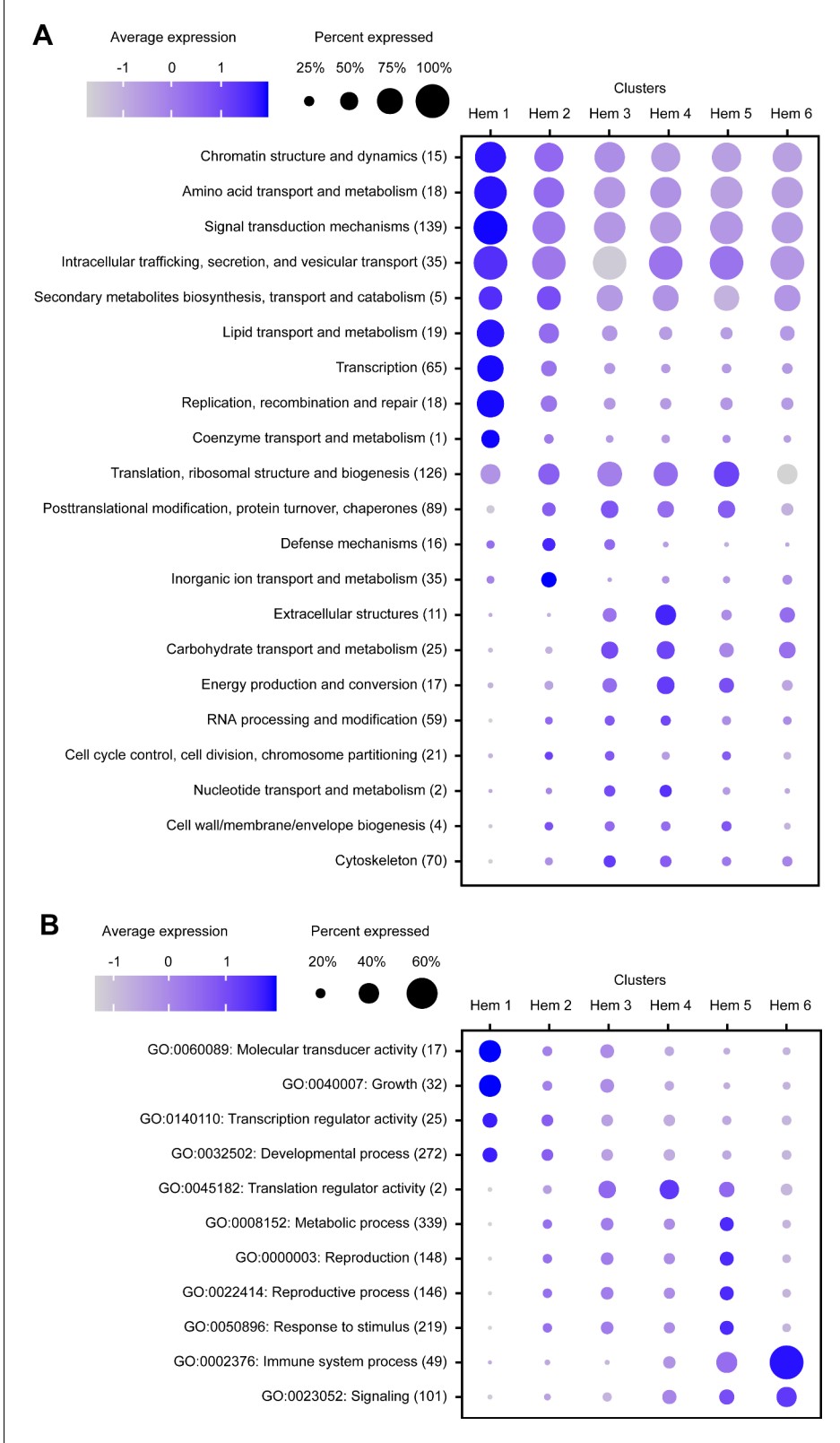

**Figure 3.** Dot plot profiling of the eukaryotic orthologous group (KOG) and gene ontology (GO) analyses in each cluster. Dot plot representing the average expression of KOGs (**A**) and GOs (**B**) per cluster. Color gradient of dots represents the expression level, while the size represents the percentage of cells expressing any genes per cluster. The numbers in parentheses represent the number of genes estimated as distinct function of KOGs or GOs.

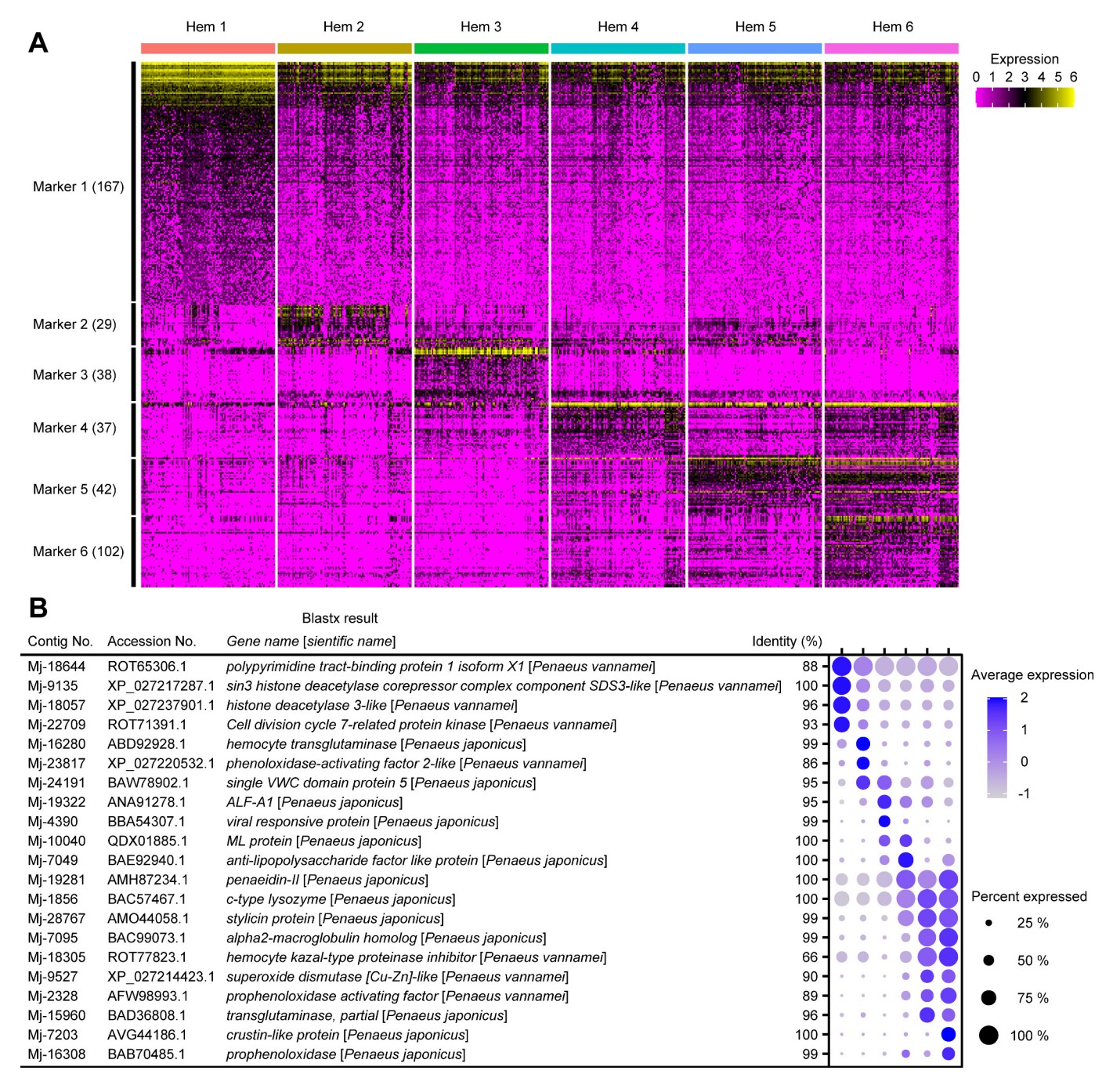

**Figure 4.** The cluster-specific marker genes predicted using the Seurat FindMarkers tool. Heat map profile of the marker in each cluster (A). Color gradient represents the expression level of each single cell. The numbers in parentheses represent the number of genes estimated as markers of each cluster. Important marker genes in each cluster (B). Color gradient of the dot represents the expression level, while the size represents the percentage of cells expressing any genes per cluster. Detailed blast results of each marker on penaeid shrimp are listed in *Supplementary file 2*.

expression of AMPs and VRP supporting Hem 3 and 4 are granular-containing hemocytes, and the expression of virus infection-related genes, VRP and ML protein, suggests that Hem 4 might play important roles when viruses infect shrimp.

In clusters Hem 5 and Hem 6, many of the cluster-specific markers showed similarity with immune-related genes of penaeid shrimp. In clusters Hem 5 and Hem 6, markers showed a high

similarity with *alpha2-macroglobulin*, *c-type lysozyme*, *stylicin*, *crustin*, *penaeidin-II*, *hemocyte kazal-type proteinase inhibitor* (*KPI*), *superoxide dismutase* (*SOD*), *prophenoloxidase* (*proPO*), *PPAF*, and *TGase*. Hem 5 and 6 were found to have high expressions of immune-related genes by GO analysis, and our maker analysis results support this observation.

## G2/M and S phase clustering and pseudo-temporal ordering of hemocytes delineate hemocyte lineages

Cell cycle of each cell was calculated by Seurat CellCycleScoring function based on the homologue genes of *Drosophila* G2/M and S phase markers (https://github.com/hbc/tinyatlas, *Kirchner and Barrera, 2019*). In the search for cell cycle-specific markers, we found that a large portion of cells in Hem 1 expressed the G2/M-related genes of *Drosophila* (*Supplementary file 3*; *Figure 5—figure supplement 2*). For examples, HP1 and Su(var)205 are known to be essential for the maintenance of the active transcription of the euchromatic genes functionally involved in cell-cycle progression, including those required for DNA replication and mitosis in *Drosophila* (*De Lucia et al., 2005*; *Paro and Hogness, 1991*). CTCF has zinc finger domains and plays an important role in the development and cell division of fly and mammalian cells (*Mohan et al., 2007*; *Rasko et al., 2001*). These findings suggested that hemocytes grouped as Hem 1 are tightly regulated by these G2/M phase-related genes to promote cell division. Among the total of 2566 cells, G1 phase consisted of 66% (1693 cells), G2/M phase consisted of 9% (236 cells), and S phase consisted of 25% cells (637 cells), respectively. With the exception of Hem 1, the percentage of G1 phase was the highest, and S phase was the second highest among all clusters; Hem 2 to Hem 6 (*Figure 5A, B*). On the other hand, Hem 1 showed the highest percentage of G2/M phase, followed by G1 phase and the lowest percentage of S phase. KOG and GO analyses of each cluster showed that Hem 1 cluster has high expression of chromatin dynamics, duplication, and growth functions (*Figure 3*). Thereby, we determined that Hem 1 is a cluster in the early stage of hemocyte development, which might have the potential of self-renewing or just came out from the hematopoietic tissue (HPT). A previous study using flow cytometry reported a significantly lower percentage of S phase, about 0.5% (*Sequeira et al., 1996*). It is possible that the genes used in the analysis of S phase had lower homology with *Drosophila* markers than the gene set used in the analysis of G2/M phase, and thus did not score as accurately as *Drosophila*.

We further performed lineage-tree reconstruction using the Monocle 3 learn_graph function to investigate the dynamics of hemocyte differentiation because the differentiation and proliferation pathways of hemocytes in shrimp and other crustaceans are still under debate (*Söderhäll, 2016*). We considered Hem 1 to be the initial state of hemocytes and set it as the starting point in the differentiation process because Hem 1 expressed cell proliferation-related genes, *TGase* (an immature hemocyte marker), and G2/M phase-related genes (*De Lucia et al., 2005*; *Paro and Hogness, 1991*; *Mohan et al., 2007*; *Rasko et al., 2001*; *Scott et al., 2007*; *Junkunlo et al., 2017*; *Junkunlo et al., 2019*). From this pseudo-temporal ordering analysis, we found two main lineages starting from Hem 1 to Hem 4, and Hem 6 at the endpoints (*Figure 5C*). In crayfish, hematopoietic stem cells are present in HPT, and two types of hemocyte lineages starting from a hematopoietic stem cell exist (*Lin and Söderhäll, 2011*). In *Penaeus monodon*, hyaline cells (i.e., agranulocytes) are considered as the young and immature hemocytes of two types of matured hemocytes (*van de Braak et al., 2002*). Our pseudo-temporal ordering analysis revealed that the hemocytes of *M. japonicus* differentiate from a single subpopulation into two major populations. The differentiation process was continuous, not discrete, which was in agreement with previous arguments on the crustacean hematopoiesis mechanism (*van de Braak et al., 2002*; *Lin and Söderhäll, 2011*; *Noonin et al., 2012*).

## Possibility of application of *Drosophila* hemocyte-type marker to shrimp

Since the hemocyte type-specific markers are better studied in *Drosophila*, we checked the Ex50 that are similar to the markers of *Drosophila* to determine whether they are applicable to shrimp. Among the specific markers of four types of *Drosophila* hemocytes, 8 genes in prohemocytes, 14 genes in plasmatocytes, 35 genes in lamellocytes, and 5 genes in crystal cells showed similarity with the shrimp genes (*Figure 6*; *Supplementary file 4*).

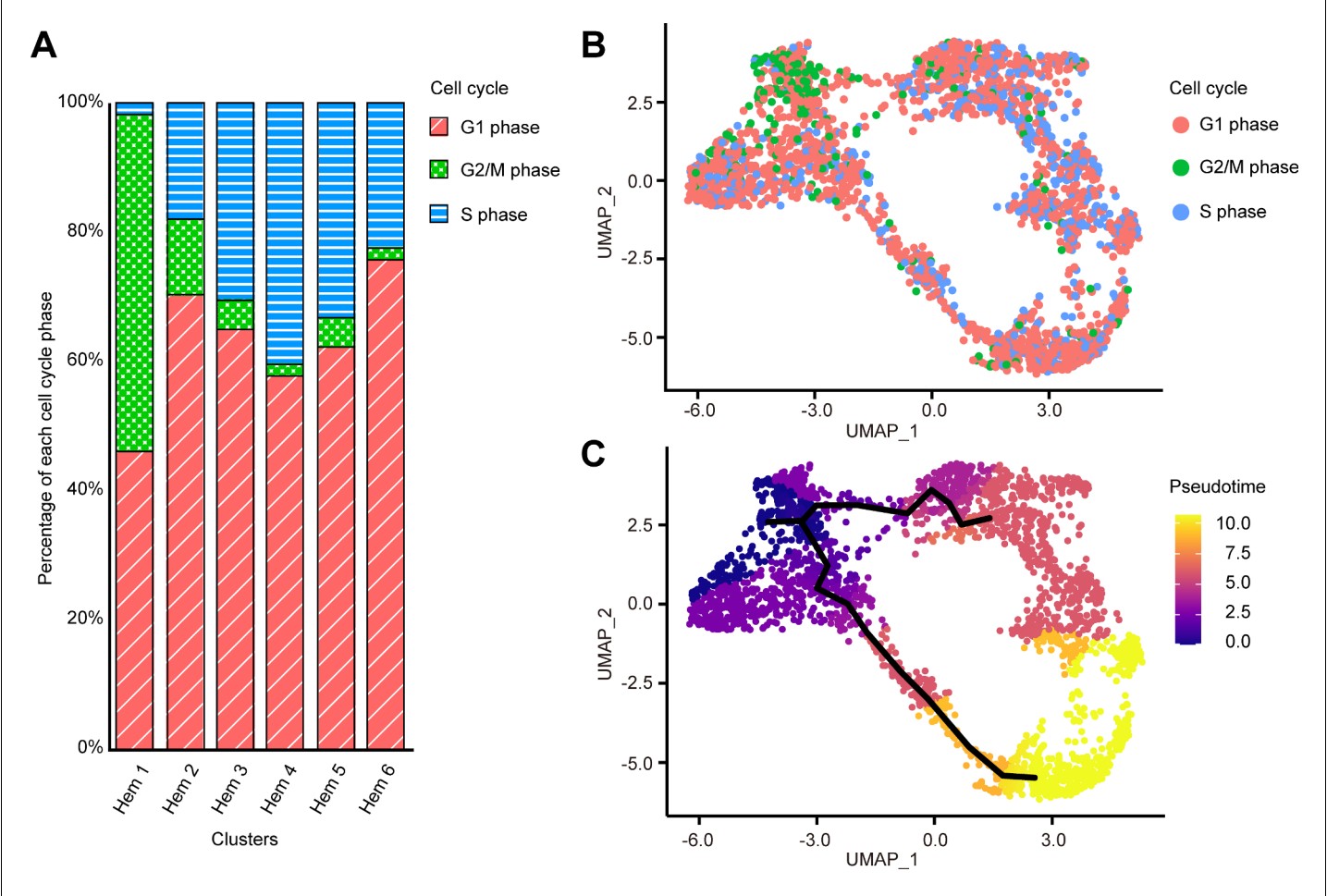

**Figure 5.** Cell cycle distribution of each cluster and pseudo-temporal ordering of hemocyte lineages. Percentage of each cell cycle on clusters (**A**). Uniform manifold approximation and projection plot of cell cycles of hemocytes from three shrimps (*n* = 2566) (**B**). Visualization of clusters onto the pseudotime map using monocle 3 (**C**). The black lines indicate the main path of the pseudotime ordering of the cells. Color gradient of each dot represents the pseudotime.

The online version of this article includes the following source data and figure supplement(s) for figure 5:

**Source data 1.** Source data of the percentage of each cell state per cluster used for *Figure 5A*.

**Figure supplement 1.** Uniform manifold approximation and projection plot of cell cycles from individual shrimps.

**Figure supplement 2.** Dot plots profiling of cell cycle-related genes in each cluster.

In case of prohemocyte markers, Mj-7095 and Mj-21032 showed cluster-specific expression and were annotated on *thioester-containing protein 4* (*Tep4*) with identity 30%, which participates in the cellular immune response to certain Gram-negative bacteria in *Drosophila* (*Figure 6*). Prohemocytes are progenitor cells in *Drosophila*; however, the expression of these marker genes was not strong in Hem 1, which was expected to be progenitor cells in shrimp, but rather was high in Hem 5 and 6. In the blastx results on penaeid shrimp proteins, Mj-7095 and Mj-21032 were annotated on *alpha2-macroglobulin* gene (*Supplementary file 1*). Alpha2-macroglobulin (a2m) of shrimp contributes clotting pathway to eliminate bacterial infection, and its proteins are contained within the secretory granules of hemocytes (*Chaikeeratisak et al., 2012*). Therefore, it is not reasonable to use these markers as a common marker with prohemocytes in shrimp.

*Drosophila* plasmatocytes are known as small round cell and professional phagocytes reminiscent of the cells from the mammalian monocyte/macrophage lineage (*Meister and Lagueux, 2003*). Mj-6 and Mj-14846 showed cluster-specific expression on the plasmatocyte markers (*Figure 6*). Mj-6 was annotated with *hemolection* (*Hml*) of *Drosophila* and *hemocytin-like* of *Litopenaeus vannamei*. Mj-

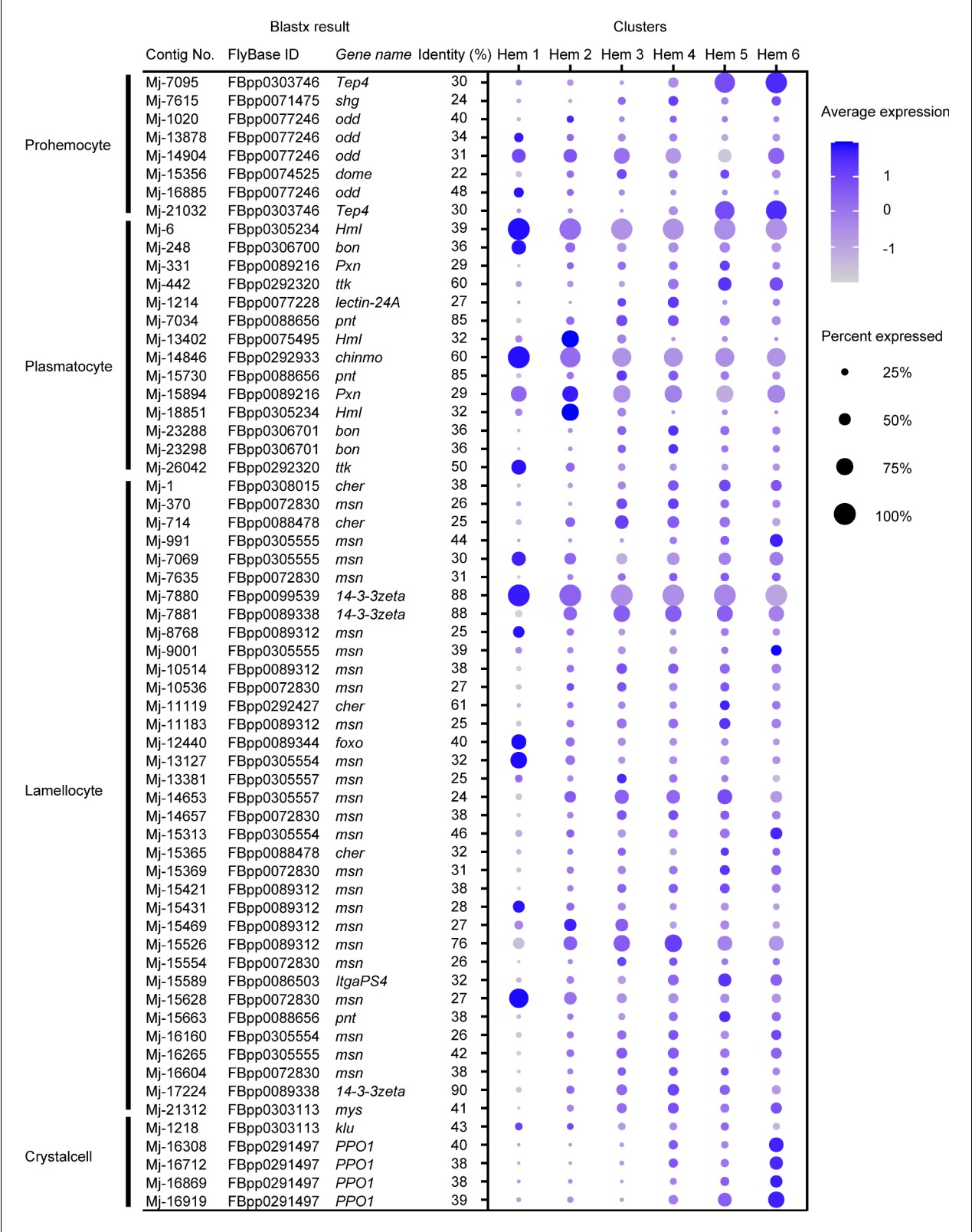

**Figure 6.** Dot plots profiling of *Drosophila* hemocyte-type markers in each cluster. Color gradient of the dot represents the expression level, while the size represents the percentage of cells expressing any gene per cluster. Detailed blast results are listed in ***Supplementary file 4***.

14846 was annotated with *chronologically inappropriate morphogenesis* (*chinmo*) of *Drosophila* and *genetic suppressor element 1-like* of *L. vannamei*. Hml of *Drosophila* is involved in the clotting reaction. It has been estimated that some of the hemocytes expressing *Hml* are self-renewing in *Drosophila* single-cell study (*Tattikota et al., 2020*). *chinmo* plays a role in the proliferation of developing hemocytes in *Drosophila* (*Flaherty et al., 2010*). Since our results also predict that Hem 1 is a group of progenitor or self-renewing cells, these genes could be a potentially useful marker for identifying progenitor or self-renewing cells of shrimp.

Mj-7880 showed characteristic expression within the identical cluster among lamellocyte markers (*Figure 6*). Mj-7880 was annotated with *14-3-3zeta* of *Drosophila* and *14-3-3-like protein* of *L. vannamei*. There are various isoforms of *14-3-3* genes, and *14-3-3zeta* is a marker for lamellocytes, but it is also known that 14-3-3 proteins function in normal cell cycle progression in *Drosophila* (*Su et al., 2001*). In shrimp, Mj-7880 is probably working as cell cycle-specific roles, which is why Mj-7880 is strongly expressed in Hem 1.

Among *Drosophila* crystal cell markers, four contigs, Mj-16308, Mj-16712, Mj-16869, and Mj-16919, were annotated on *prophenoloxidase 1* (*PPO1*) of *Drosophila* (*Figure 6*). These four contigs were also annotated on *proPO* of *M. japonicus* and *phenoloxidase 3-like* of *L. vannamei*. From the single-cell study of *Drosophila* hemocytes, the expression level of *PPO1* increased with the maturation of crystal cells (*Tattikota et al., 2020*). Crystal cells contain about −5% of total number of hemocytes of *Drosophila* and contain the enzymes necessary for humoral melanization that accompanies a number of immune reactions (*Meister and Lagueux, 2003*). It is very interesting to note that the ratio of Hem 6 is larger in shrimp compared to in *Drosophila* crystal cells, but the expression of *proPO* is highest in Hem 6, the estimated endpoint of differentiation by pseudotime analysis (*Figure 5C*). It is not possible to assign crystal cells of *Drosophila* and Hem 6 of shrimp as homologous cells because of the unknown similarities in morphology and other functions, but at least *PPO/proPO* can be considered to be a marker gene commonly expressed in mature hemocytes of both *Drosophila* and shrimp.

## Expression of cell growth-related genes

The exploration of cluster-specific markers revealed that cell growth-related genes were specifically expressed in certain clusters. We identified 10 cell growth-related genes from Ex50 that are predicted to be involved in cell growth and differentiation: *extracellular signal-regulated kinase* (*ERK*), *mitogen-activated protein kinase kinase 4* (*MKK4*), *insulin-like growth factor-binding protein 4* (*IGFBP-4*), *vascular endothelial growth factor 1* (*VEGF-1*), *VEGF-3*, *astakine*, *crustacean hematopoietic factor-like protein* (*CHF-like*), *PDGF/VEGF-related factor 1*, *growth hormone secretagogue receptor type 1* (*GHSR*), and *platelet-derived growth factor receptor alpha-like* (*PDGFRA-like*) (*Figure 7A*; *Supplementary file 1*).

Both *ERK* and *MKK4* are a series of genes involved in mitogen-activated protein kinase (MAPK) pathways. In *Drosophila,* MAPK pathway is activated by signaling from several receptors, causing hemocyte proliferation and differentiation (*Zettervall et al., 2004*). Although it has been investigated that MAPK pathway contributes to the defense responses of shrimp (*Yang et al., 2012*; *He et al., 2018*; *Luo et al., 2020*), it is not yet known whether this pathway directly affects the proliferation or differentiation of hemocytes. However, the high expression of *ERK* and *MKK4* in Hem 1 strongly suggests that MAPK pathway is involved in the proliferation of hemocytes in shrimp. Therefore, not only immune responses but also cell-type variation is a good point of focus for future research on studies of MAPK pathway in shrimp.

Hem 2 highly expressed *IGFBP-4* (*Figure 7*). IGFBP delivers IGFs to the target cells in mammal studies and is essential for cell growth or differentiation (*Varma Shrivastav et al., 2020*). The high expression of a receptor of the insulin-like peptide at mature hemocytes in the mosquito suggests that the insulin signaling pathway regulates hemocyte proliferation (*Castillo et al., 2011*). The silencing and overexpression of IGFBP caused a decrease and increase in the growth of hemocytes of abalone *Haliotis diversicolor* (*Wang et al., 2015*). These studies indicate that the IGFBP-related insulin signaling pathway is important for hemocyte proliferation and differentiation in invertebrates. IGFBP-4 might play an essential role in the differentiation of hemocytes from Hem 1 to Hem 2 in shrimp. The high expression of *IGFBP* is determined in the brain and gonads of *L. vannamei* (*Ventura-López et al., 2017*). This fact also suggests that IGFBP plays a possible role in organ growth and maturation in shrimp. It is worth noting that the expression of *IGFBP-4* is strong at the site

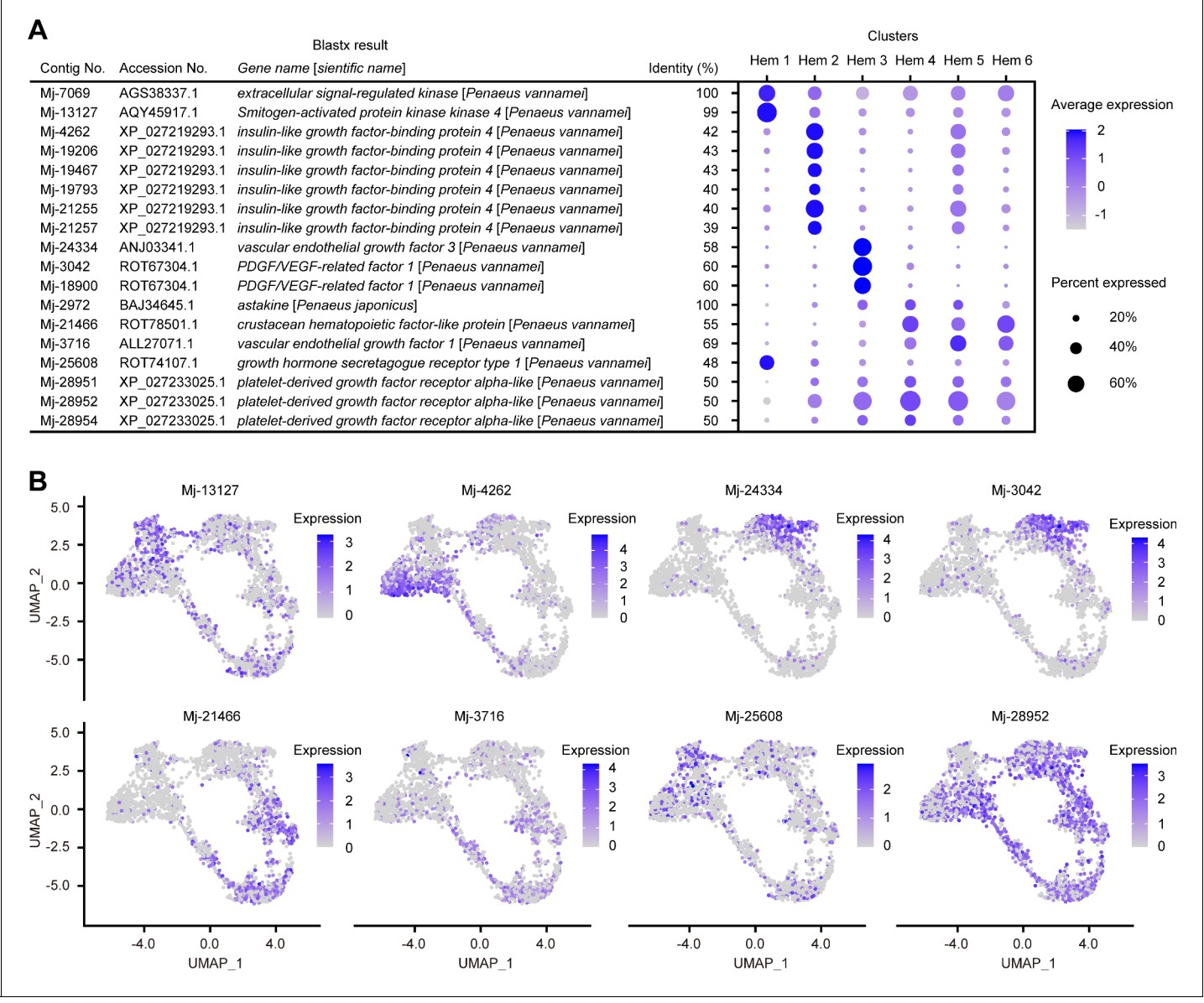

**Figure 7.** Cell growth-related gene expressions on clusters and single hemocytes. Dot plots profiling of cell growth-related genes in each cluster (**A**). Color gradient of the dot represents the expression level, while the size represents the percentage of cells expressing any gene per cluster. Expression profiling of cell growth-related genes on uniform manifold approximation and projection plot (**B**). Color gradient of each dot represents the expression level. The details of the identified genes are listed in *Supplementary file 1*.

where Hem 3 and Hem 5 seem to be in the early stage of differentiation, which may indicate that IGFBP-4 is acting to initiate differentiation into Hem 3 and Hem 5 from Hem 2 (*Figure 7B*).

Cells expressing *VEGF-3* and *PDGF/VEGF-related factor* 1 were dominant in Hem 3. The vascular endothelial growth factor (VEGF) signaling pathway is essential for vasculogenesis, cell proliferation, and tumor migration in mammals (*Neufeld et al., 1999*; *Peichev et al., 2000*). Furthermore, in *Drosophila*, VEGF homologs control the number of circulating hemocytes (*Munier et al., 2002*). The high expression of *VEGF-3* and *PDGF/VEGF-related factor* 1 in Hem 3 indicated that Hem 2 would differentiate into Hem 3 the VEGF signaling pathway. Interestingly, a different homologue of VEGF, *VEGF 1*, was highly expressed in Hem 4, Hem 5, and Hem 6. These hemocytes are estimated as end-points of differentiation by pseudotime analysis (*Figure 5C*). This means that the different types of VEGFs, *VEGF-1* and *VEGF-3*, may induce differentiation into different hemocytes.

*Astakines*, which were expressed in Hem 3, Hem 4, and Hem 5, are produced by granular-containing hemocytes and are released into the plasma (*Söderhäll et al., 2005*). Astakines function as decreasing the extracellular TGase activity and inducing the differentiation of hemocyte precursor into maturation (*Söderhäll, 2016*). Our analysis showed that *astakine* was expressed from a part of matured hemocytes, which is predicted as granular-containing hemocytes. Therefore, as in the previous study (*Lin et al., 2011*), it is thought that *astakine* act on precursor hemocytes, Hem 1 and Hem 2, to promote maturation in shrimp.

From Ex50 genes, we identified two candidate receptors related to cell growth, *GHSR* and *PDGFRA-like*. GHSR is a receptor for ghrelin and stimulates growth hormone secretion in vertebrates (*Smith et al., 2001*). Growth hormone-releasing peptide-6 (GHRP-6) is a synthetic peptide that mimics the effect of the endogenous ligand ghrelin and has a strong growth hormone secretagogues activity. In shrimp, GHRP-6 functions as a growth promoter peptide. More interestingly, the total hemocyte number in hemolymph was increased by supplying of GHRP-6 (*Martínez et al., 2017*). Therefore, GHSR is thought to play an important role in increasing hemocyte number in shrimp. PDGFRA, as the name suggests, is the receptor for platelet-derived growth factor (PDGF). It regulates cellular growth and differentiation by binding to its ligand (*Lei and Kazlauskas, 2009*). In crayfish, PDGF regulates hemocyte differentiation from immature to matured (*Junkunlo et al., 2017*). The strong expression of *PDGFRA* in shrimp was detected in Hem 2 to Hem 6, not in Hem 1, suggesting that PDGF signals are important for hemocyte maturation but not for proliferation of hemocytes in shrimp.

So far, we described the expression of proliferation- and differentiation-promoting genes. However, there was also the specific expression of the hemocyte homeostasis regulatory gene, *crustacean hematopoietic factor* (*CHF*) (*Lin and Söderhäll, 2011*). CHF is a hematopoietic factor of crayfish, and the silencing of CHF leads to an increase in the apoptosis of cells in HPT and a reduction in the number of circulating hemocytes (*Jiao et al., 2016*). Additionally, the silencing of laminin, a receptor of CHF, reduces the number of circulating hemocytes by decreasing the number of agranulocytes, as opposed to granulocytes, in *P. vannamei* (*Charoensapsri et al., 2015*). *CHF-like* was expressed in cluster Hem 4, Hem 5, and Hem 6 (*Figure 7*). Taken together, *CHF-like* expressed from matured hemocytes might work as a hematopoietic factor against immature hemocytes, such as Hem 1 and Hem 2.

## Expression of immune-related genes

Hemocytes of shrimp play key roles in their immunity; therefore, we identified immune-related genes from Ex50, and then analyzed their expression levels to deduce the detailed immune functions of each cluster (*Supplementary file 1*).

AMPs and antimicrobial enzymes play the most important role in the immunity of shrimps and are well known to be stored in granulocytes (*Bachère et al., 2004*; *Rosa and Barracco, 2010*). The expression patterns of AMPs and antimicrobial enzymes revealed that the major AMPs of penaeid shrimp were expressed in clusters Hem 4 to Hem 6 (*Figure 8A, B*, *Figure 8—figure supplement 1*). Therefore, Hem 4 to Hem 6 were predicted to be granulocytes. Whereas almost all the expressions of AMP genes were increased as hemocytes grew (*Figure 5C*, *Figure 8B*), the expression patterns of *ALFs* were different from other AMPs (*Figure 8—figure supplement 1*). It is predicted that the diversity of shrimp *ALFs* contributes to create synergism-improving shrimp antimicrobial defenses (*Rosa et al., 2013*). Analysis of each function of *ALFs* may lead to a more detailed understanding of immune function in each cluster of hemocytes in shrimp.

Clotting is an important innate immune response in shrimp. Transglutaminase (TGase) plays critical roles in clotting and hematopoiesis, and there are two types of TGases in shrimp (*Lin et al., 2008*; *Söderhäll, 2016*; *Lin and Söderhäll, 2011*; *Chen et al., 2005*). In crayfish, different types of TGases are localized in different types of hemocyte, in immature hemocytes and granular hemocytes, respectively (*Junkunlo et al., 2020*). In our scRNA-seq, different types of TGases were expressed in different cluster of hemocytes (*Figure 8—figure supplement 2*), indicating that shrimp TGases are also expressed in different places. More interestingly, it is reported that TGase is not expressed in all granulocytes in crayfish (*Junkunlo et al., 2020*). Our result was in accordance with the previous study because not 100% of the hemocytes expressed TGase in clusters Hem 5 and Hem 6. Alpha2-a2m of shrimp contributes to clotting pathway and are stored in the granules of hemocytes (*Chaikeeratisak et al., 2012*). Since our results also showed strong expression of a2m in

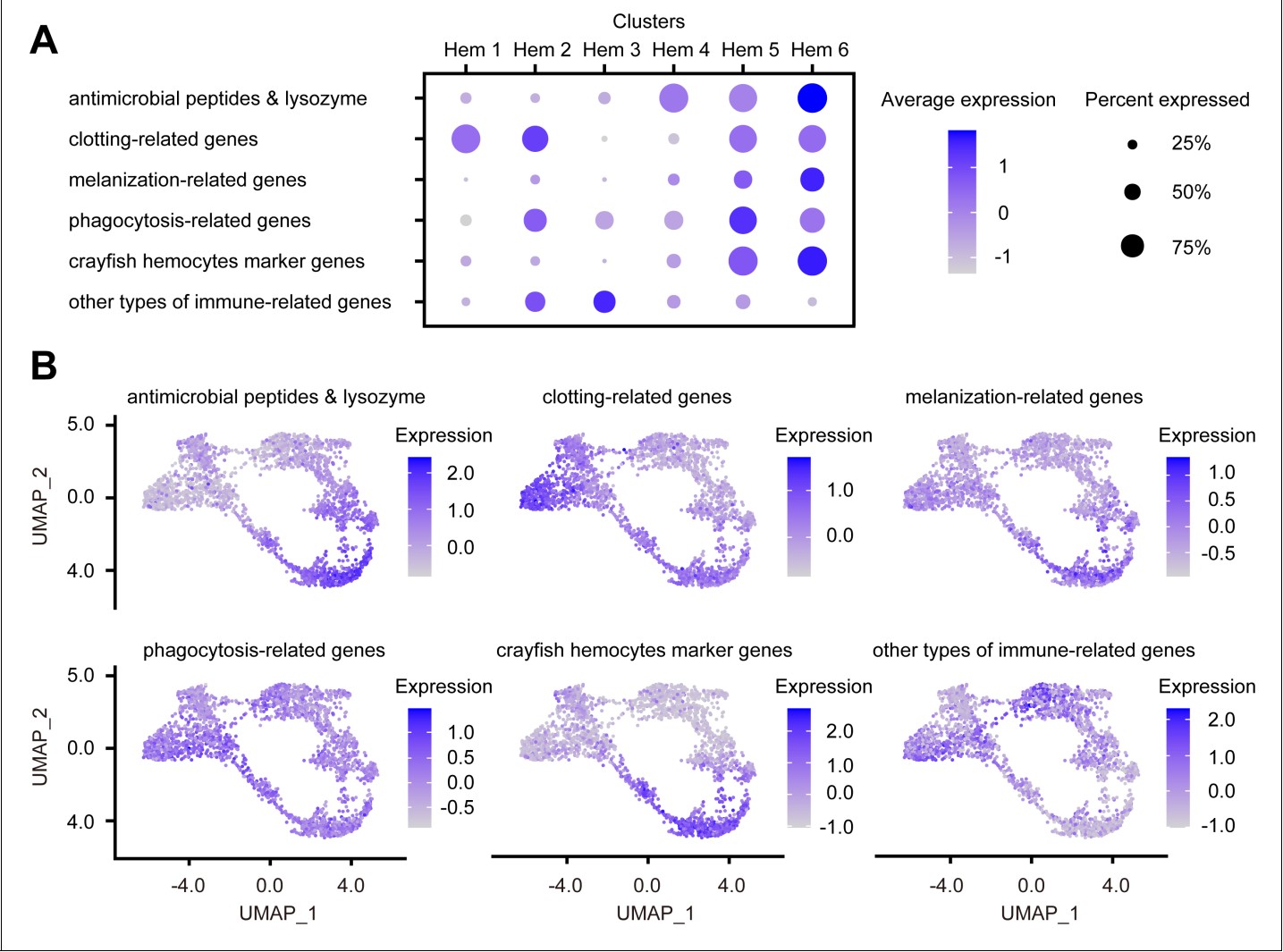

**Figure 8.** Dot plots and uniform manifold approximation and projection (UMAP) plot profiling of immune-related genes. Dot plot representing the average expression of each immune-related gene per cluster (**A**). Color gradient of the dot represents the expression level, while the size represents the percentage of cells expressing any genes per cluster. The numbers in parentheses represents the number of genes estimated as immune-related. Expression profiling of immune-related genes on UMAP plot (**B**). Color gradient of each plot represents the expression level. The details of the identified genes are listed in ***Supplementary file 1***.

The online version of this article includes the following figure supplement(s) for figure 8:

**Figure supplement 1.** Dot plot representing the antibacterial peptides and lysozyme-related genes per cluster based on average expression.
**Figure supplement 2.** Dot plot representing the clotting-related genes per cluster based on average expression.
**Figure supplement 3.** Dot plot representing the melanization-related genes per cluster based on average expression.
**Figure supplement 4.** Dot plot representing the phagocytosis-related genes per cluster based on average expression.
**Figure supplement 5.** Dot plot representing the crayfish hemocyte marker genes per cluster based on average expression.
**Figure supplement 6.** Dot plot representing the other types of immune-related genes per cluster based on average expression.

Hem 5 and Hem 6, these clusters are predicted to be a granulocyte. Coagulation factor and hemocytin are related to clotting pathway in invertebrates (***Lavine and Strand, 2002***; ***Kotani et al., 1995***). Strong expression of the whole clotting-related genes was predicted in Hem 1, Hem 2, Hem 5, and Hem 6, except Hem 3 and Hem 4 (***Figure 8***). Although it is not possible to estimate which genes are working in clotting pathway and how, it is clear that there were populations, Hem 3 and Hem 4, that are not involved in clotting.

Melanization is performed by PO and controlled by the prophenoloxidase (proPO) activation cascade in shrimp (***Amparyup et al., 2013***). Except prophenoloxidase-activating factor 2-like (PPAF2-

like) and QM protein, melanization-related genes were expressed in Hem 4 to Hem 6. This result is consistent with the fact that *proPO* was a marker gene for mature hemocytes (*Figure 8—figure supplement 3*). In the previous study, the strong PO activity was observed in granules of hemocytes in crayfish (*Giulianini et al., 2007*), suggesting that PO activity and granules are closely related. In addition, it is considered that mature granulocytes are involved in melanization. Although the detailed functions of PPAF2-like and QM protein were still unknown except that these genes are involved in proPO cascade in shrimp, it is reasonable to conclude that melanization or proPO cascade is the product of a cascade reaction through the entire hemocytes, not just granulocytes.

Phagocytosis is a crucial defense mechanism against bacteria and viruses in shrimp. Many types of genes are involved in phagocytosis, such as receptors for recognition, downstream signal pathways, and intracellular regulators (*Liu et al., 2020*). In kuruma shrimp *M. japonicus*, *integrin alpha PS* is reported as a marker for hemocytes that mediates phagocytosis (*Koiwai et al., 2018*). However, it is reported that different subpopulations of hemocytes seem to exhibit specific preferences of different bacteria or viruses in phagocytosis (*Liu et al., 2020*). Since there are many cascades for phagocytosis and various molecules work together, it is difficult to determine which cluster is centralized, but various phagocytosis-related genes are expressed relatively strongly in Hem 5 (*Figure 8—figure supplement 4*). *Rab7* was found to be highly expressed in Hem 1, which is known to be involved in the endosomal trafficking pathway of virus-infected penaeid shrimp (*Zhao et al., 2015*). However, there have been no detailed functional studies of *Rab7* on whether it is involved in phagocytosis in shrimp, so it is early to determine its role in Hem 1.

Some of the hemocyte type-specific markers related to their immune function have been studied in crayfish: proPO in matured hemocytes, copper/zinc superoxide dismutase (SOD) in semi-granular cells (SGC), and Kazal-type proteinase inhibitor (KPI) in granular cells (GC), and TGase in immature cells (*Söderhäll, 2013*). The usefulness of proPO and TGase as markers has already been discussed above. Both *KPI* and *SOD* were mostly expressed at both Hem 5 and Hem 6, and their expression levels were similar between these clusters (*Figure 8—figure supplement 5*). These results suggest that the functional segregation of hemocytes in shrimp is different from that in crayfish.

We also identified other immune-related genes (*Figure 8—figure supplement 6*). *Notch* and *pellino* regulate AMPs expression through the signaling pathway (*Zhang et al., 2020*; *Li et al., 2014*; *Ning et al., 2018*). As discussed in the section explaining cluster-specific markers, *ML protein, VRP,* and *Vago 5* are related to virus infection. Molecules that directly affect defense, such as AMPs and clotting-related genes, are more abundantly expressed in Hem 4 to Hem 6, whereas molecules that regulate immune system seem to be more abundantly expressed in Hem 1 to Hem 3. In other words, it was predicted that agranulocytes play a role in signaling for immune system, and that granulocytes are responsible for direct biological defense functions. In addition, the expression of virus infection-related genes, *ML protein*, *VRP,* and *Vago 5*, suggests Hem 2, Hem 3, and Hem 4 might play important roles when virus infect shrimp.

In the mosquito, scRNA-seq revealed a new subpopulation called 'antimicrobial granulocytes' that expressed characteristic AMPs (*Raddi et al., 2020*). Similarly, in *M. japonicus*, the expression patterns of immune-related genes were also different among certain clusters, suggesting that shrimp hemocytes are more heterogeneous than previously thought. It is anticipated that the class of granulocytes discussed in previous studies is actually a mixture of clusters exhibiting different roles.

## Validation of marker genes and the relationship between clusters and morphology

Our scRNA-seq results revealed six major subpopulations and their marker genes, and the possible differentiation trajectory of kuruma shrimp *M. japonicus* hemocytes. Next, we examined the correlation between the morphology and expression of marker genes. Two major populations of hemocytes were sorted based on the forward versus side scatter plot obtained using microfluidic-based fluorescence-activated cell sorter (FACS). The sorted populations were observed using microscopy. FACS was able to separate hemocytes into two morphologically different populations (*Figure 9A–E*, *Figure 9—figure supplement 1*): smaller cells with low internal complexity in region 1 (R1) (50.7 ± 5.2%) and larger cells with high internal complexity in region 2 (R2) (47.7 ± 3.4%). From differential interference contrast (DIC) and dye staining imaging (*Figure 9A–D*), we observed that cells in the R1 region contained no or few granules in the cytoplasm. The nucleus occupied a large portion of the

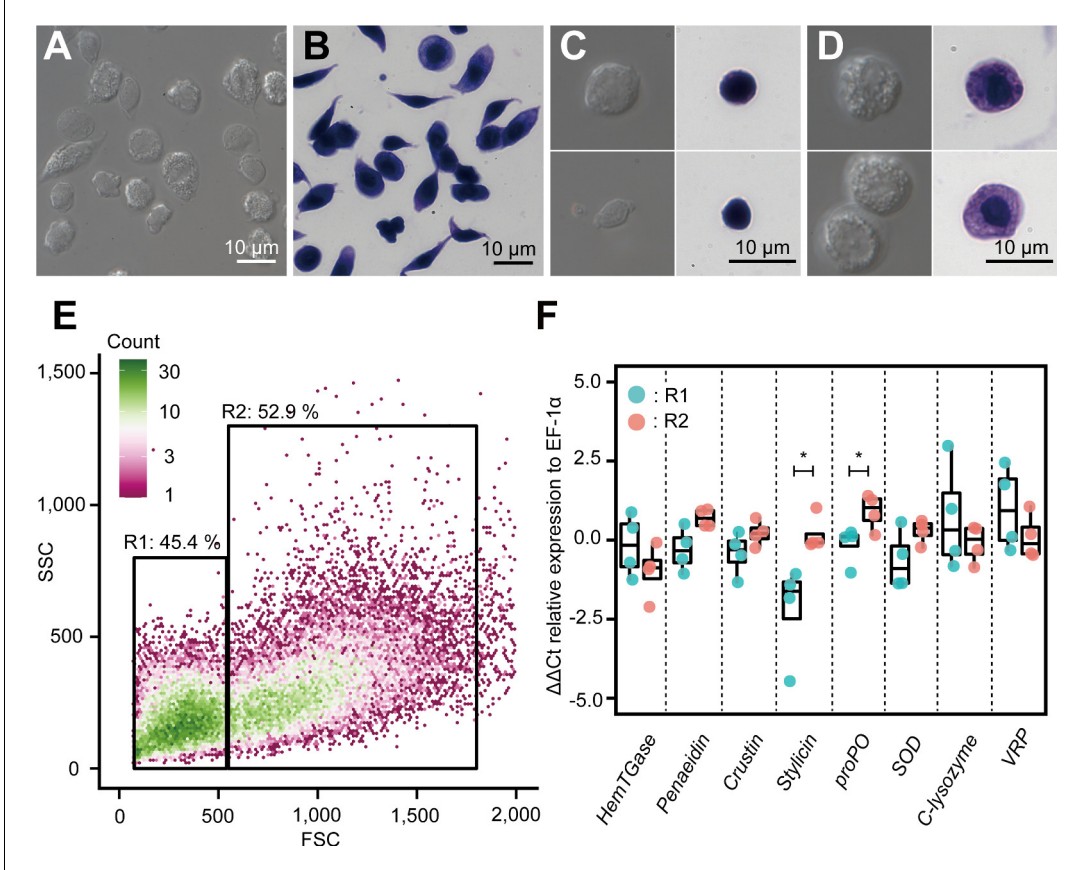

**Figure 9.** Morphological analysis of hemocytes and transcript profiles based on morphology. (**A**) Differential interference contrast (DIC) image of unsorted total hemocytes. (**B**) Dye-stained total hemocytes. (**C**) DIC imaging and dye staining of region 1 (R1)-sorted hemocytes. (**D**) DIC imaging and dye staining of region 2 (R2)-sorted hemocytes. (**E**) Fluorescence-activated cell sorting (FACS) analysis of hemocytes. Based on the forward scatter (FSC) and side scatter (SSC) two-dimensional space, two regions (R1 and R2) were obtained. (**F**) Differential gene expression analysis between R1 and R2 of hemocytes sorted using FACS. ΔΔCt values were analyzed using qRT-PCR. Higher ΔΔCt values indicate a higher accumulation of mRNA transcripts. The p values shown in the figures are represented by *p<0.05.

The online version of this article includes the following source data and figure supplement(s) for figure 9:

**Source data 1.** Source data for CT values of each gene used for *Figure 9F*.

**Figure supplement 1.** Ffluorescence-activated cell sorting analysis of hemocytes from four individual shrimps (**A–D**).

volume in these cells (*Figure 9C*). Conversely, those in the R2 region had many granules in the cytoplasmic region, which occupied a large portion of cells (*Figure 9D*).

We conducted qRT-PCR analysis to determine the expression of some representative genes in these populations. The results showed that the average ΔΔCt values of the transcripts of *HemTGase* were higher in R1 hemocytes than in R2 hemocytes, while the values of *penaeidin*, *crustin*, *stylicin*, *proPO*, and *SOD* were higher in R2 hemocytes (*Figure 9F*). The average ΔΔCt values of transcripts of *c-type lysozyme* and *VRP* were similar in R1 and R2 hemocytes. The combination of FACS and qRT-PCR results confirmed that the gene expression of the two populations in shrimp hemocytes was roughly divided based on morphology, that is, agranulocytes and granulocytes, which is consistent with our scRNA-seq analysis. Some of these genes were not statistically different in the t-test, but due to the small sample size, we used the mean values for discussion. Granulocytes found in the R2 region expressed major AMPs, such as *penaeidin*, *crustin*, and *stylicin*, suggesting that they consist of clusters Hem 4 to Hem 6. However, *c-type lysozyme*, which was expressed in Hem 4 to Hem 6 in scRNA-seq, was higher on the cells in R1 region. The reason why only *c-lysozyme* was expressed in different population from the other AMPs was unclear. *VRP*, which is a marker of Hem 3 (*Figure 8—figure supplement 6*), was expressed in cells in both R1 and R2 regions, indicating that Hem

3 exists in both populations and is indistinguishable from the morphological characteristics. Our conclusion that the cells in R1 and R2 regions separated by FACS consist of Hem 1–3 and Hem 4–6, respectively, is quantitatively supported by the fact that Hem 1–3 and Hem 4–6 cover 60.4 and 39.6% of whole cells, respectively.

## Discussion

Our single-cell transcriptome analysis revealed that there are six subpopulations of hemocytes in shrimp. This result provides us a detailed and advanced understanding of the cell classification over previous strategies based on various approaches, such as simple staining (*Söderhäll and Smith, 1983*; *Johansson et al., 2000*; *van de Braak et al., 2002*), monoclonal antibodies (*Xing et al., 2017*), flow cytometry (*Koiwai et al., 2017*), and lectin-binding profiles (*Koiwai et al., 2019*). We hope that future studies will clarify the relationships between these phenotypic features and our classification to comprehend the role of each cluster in the immune system of shrimp.

The cluster-specific markers and cell proliferation-related genes found here helped us to understand how shrimp hemocytes differentiate. A strong expression of *TGase*, cell proliferation-related genes, and G2/M state-related genes in Hem 1 suggested that hemocytes in this cluster are oligopotent and located upstream in the differentiation process. In crustaceans, especially shrimp and crayfish, it is known that hemocytes are produced in HPT, and that differentiated hematopoietic cells from HPT circulate in the body fluid (*van de Braak et al., 2002*; *Söderhäll, 2016*; *Söderhäll, 2013*). Hem 1 accounts for only 6.5% of the analyzed cells, which indicates that only a very small fraction of oligopotent or initial state hemocytes exists among the circulating hemocytes. Many of the markers that were characteristically expressed in Hem 1 could not be functionally predicted by the BLAST search (*Supplementary file 1*). We speculate that the characteristic genes of Hem 1 are associated with cell duplication or differentiation. The division and differentiation mechanisms of shrimp hemocytes are still largely unknown, and no techniques on culture shrimp hemocytes in vitro have been reported. The analysis of these unknown gene characteristics may reveal these mechanisms.

Our results also revealed that the growth-related genes were expressed on specific clusters (*Figure 7*). It was predicted that several growth factors are involved in differentiation and maturation of hemocytes, and that these factors play roles in the maturation of hemocytes through several cellular processes. In *P. monodon*, a model has been proposed in which hemocytes differentiate linear to two types of granulocytes from a single type of agranulocytes (*van de Braak et al., 2002*). Our study strongly supports this previous study. We would also like to emphasize that shrimp hemocytes are not completely separated into distinct cell types, and their differentiation process is rather continuous. Loss and gain function studies of these growth-related genes are necessary to prove the full differentiation process of penaeid shrimp hemocytes.

Crayfish's hemocytes and *Drosophila* hemocytes have been classified into three and four major types according to the marker genes, respectively (*Tattikota et al., 2020*; *Cho et al., 2020*; *Söderhäll, 2016*). The relationship between *Drosophila* and shrimp was almost impossible to establish on the hemocyte-type markers. Also, for crayfish hemocyte markers, there was no difference between clusters according to *SOD* and *KPI* that are the markers of SGC and GC, respectively. Insects and crustaceans are thought to become independent about 500 million years ago (*Rota-Stabelli et al., 2013*; *Thomas et al., 2020*), and shrimp and crayfish are thought to have become evolutionarily independent about 450 million years ago (*Wolfe et al., 1901*). Thus, it is straightforward to reason that the functions of these genes have changed during the evolutionary process. Therefore, we should not simply argue that the morphological and functional similarities between shrimp and *Drosophila*/crayfish hemocytes are the same. Interestingly, however, *proPO* (or *PPO*) was found to be highly expressed in mature hemocytes in shrimp, *Drosophila*, and crayfish alike. Therefore, it is possible that *proPO* is a marker gene for mature hemocytes, at least in common with these three species. It is unclear to what extent this trend will hold, but it is interesting to examine why this trend is conserved across species.

There have been a number of studies to investigate the function of hemocytes, but most of them were done on whole hemocytes because of the difficulty in distinguishing and isolating cell types. Here, we predicted the major functions, that is AMPs releasing, clotting, melanization, and phagocytosis, of each hemocyte population. The strong expression of the AMPs, melanization, and phagocytosis-related genes in mature hemocytes revealed that granulocytes are the cells that play a central

role in immune activity in shrimp. Furthermore, clotting-related genes were strongly expressed in Hem 1 and Hem 2, indicating that agranulocytes are the main player in clotting. In addition, the expression of genes that is known to be elevated by viral infection was high in Hem 3 and Hem 4. This may indicate that these hemocyte populations are the ones that work during virus infection. Or, the percentage of this hemocyte population may increase during viral infection, while the amount of gene expression in the hemocytes remains the same. This aspect will be clarified by studying the expression dynamics in shrimps infected with viruses or bacteria. In mature hemocytes, the final products of the immune response, such as *AMPs*, *a2m*, and *proPO*, were strongly expressed, whereas the genes responsible for the middle of the cascade, such as *PPAF*, *Notch*, *pellino*, *ML protein*, and *Vago 5*, were expressed in immature or differentiated hemocytes. This observation suggests that the immune system is not governed solely by granulocytes, but there is a communication with agranulocytes in response to foreign substances.

In conclusion, we succeeded in classifying shrimp hemocytes into six subpopulations based on their transcriptional profiles, while they were only classified into two groups using FACS. Furthermore, our results imply that hemocytes differentiate from a single initial population. Although we have not yet successfully cultured crustacean hemocytes in passaging cultures, information on these subpopulations and marker genes will provide a foothold for hemocyte culture studies. Despite our success in the classification of hemocytes, we have not yet been able to fully understand the functions of each hemocyte group in detail. One reason for this is that the functions of some marker genes are still unknown. The present single-cell transcriptome data serves as a platform providing the necessary information for the continuous study of shrimp genes and their functions. Additionally, we have only determined the subpopulations of hemocytes in the normal state, so that our next goal will be to analyze the hemocytes in the infected state of certain diseases at different levels of cell maturation. In this way, we will be able to identify the major subpopulations that may work against the infectious agent. Likewise, it will be interesting to see how hemocytes from fertilized eggs mature. Additionally, single-cell analysis of HPTs can also be expected to reveal more detailed differentiation mechanisms. Unlike terrestrial invertebrates, such as *Drosophila* and mosquitoes, shrimps are creatures that live in the ocean and are evolutionarily distant from each other. There is room for improvement in genomic information, library preparation efficiency, functional prediction, etc., but this kind of single-cell study in shrimp will provide efficient solutions to aquaculture problems.

# Materials and methods

## Key resources table

| Reagent type (species) or resource | Designation | Source or reference | Identifiers | Additional information |
|---|---|---|---|---|
| Biological sample (*Marsupenaeus japonicus*) | Hemocytes | NA | NA | Hemocytes from hemolymph from 20 g of kuruma shrimp |
| Sequence-based reagent | 1st PCR primer | *Macosko et al., 2015* | DOI: 10.1016/j.cell.2015.05.002 | AAGCAGTGGTATCAACGCAGAGT |
| Sequence-based reagent | P5 universal primer | Illumina, Inc | NA | AATGATACGGCGACCACCGAGA TCTACACGCCTGTCCGCGGAAG CAGTGGTATCAACGCAGAGT*A*C |
| Sequence-based reagent | i7 index primer | Illumina, Inc | NA | N703: CAAGCAGAAGACGGCATACG AGATTTCTGCCTGTCTCGTGGGCTCGG N704: CAAGCAGAAGACGGCATACGAG ATGCTCAGGAGTCTCGTGGGCTCGG N705: CAAGCAGAAGACGGCATACGAGATAG GAGTCCGTCTCGTGGGCTCGG |
| Sequence-based reagent | Custom sequence primer | *Macosko et al., 2015* | DOI: 10.1016/j.cell.2015.05.002 | GCCTGTCCGCGGAAGCAG TGGTATCAACGCAGAGTAC |

*Continued on next page*

*Continued*

| Reagent type (species) or resource | Designation | Source or reference | Identifiers | Additional information |
|---|---|---|---|---|
| Sequence-based reagent | EF-1α | *Koiwai et al., 2019* | DOI:10.1007/s12562-019-01311-5 sequence accession number: AB458256 | For: ATTGCCACACCGCTCACA Rev: TCGATCTTGGTCAGCAGTTCA |
| Sequence-based reagent | HemTGase | *Yeh et al., 2006* | DOI: 10.1016/j.bbapap.2006.04.005 sequence accession number: DQ436474 | For: GAGTCAGAAGTCGCCGAGTGT Rev: TGGCTCAGCAGGTCGTTTAA |
| Sequence-based reagent | Penaeidin | *An et al., 2016* | DOI: 10.1016/j.dci.2016.02.001 sequence accession number: KU057370 | For: TTAGCCTTACTCTGTCAAGTGTACGCC Rev: AACCTGAAGTTCCGTAGGAGCCA |
| Sequence-based reagent | Crustin | *Hipolito et al., 2014* | DOI: 10.1016/j.dci.2014.06.001 sequence accession number: AB121740 | For: AACTACTGCTGCGAAAGGTCTCA Rev: GGCAGTCCAGTGGCTTGGTA |
| Sequence-based reagent | Stylicin | *Liu et al., 2015* | DOI: 10.1016/j.fsi.2015.09.044 sequence accession number: KR063277 | For: GGCTCTTCCTTTTCACCTG Rev: GTCGGGCATTCTTCATCC |
| Sequence-based reagent | proPO | *Koiwai et al., 2019* | DOI: 10.1007/s12562-019-01311-5 sequence accession number: AB073223 | For: CCGAGTTTTGTGGAGGTGTT Rev: GAGAACTCCAGTCCGTGCTC |
| Sequence-based reagent | SOD | *Hung et al., 2014* | DOI: 10.1016/j.fsi.2014.07.030 sequence accession number: AB908996 | For: GCCGACACTTCCGACATCA Rev: TTTTGCTTCCGGGTTGGA |
| Sequence-based reagent | C-lysozyme | *Hikima et al., 2003* | DOI:10.1016/s0378-1119(03)00761-3 | For: ATTACGGCCGCTCTGAGGTGC Rev: CCAGCAATCGGCCATGTAGC |
| Sequence-based reagent | VRP | *Elbahnaswy et al., 2017* | DOI: 10.1016/j.fsi.2017.09.045 sequence accession number: LC179543 | For: CTACGGTCGCTACCTTCGTTTG Rev: TCAACAACGCTTCTGAACTTATTCC |
| Commercial assay or kit | TRI REAGENT | Molecular Research Center, Inc | TR118 | NA |
| Commercial assay or kit | Direct-zol RNA MiniPrep | Zymo Research | R2050 | NA |
| Commercial assay or kit | Dynabeads Oligo(dT)25 | Thermo Fisher Scientific | DB61002 | NA |
| Commercial assay or kit | Direct RNA Sequencing kit | Oxford Nanopore Technologies | SQK-RNA002 kit | NA |
| Commercial assay or kit | MinION Flow Cell | Oxford Nanopore Technologies | Flow Cell R9.4.1 | NA |
| Commercial assay or kit | Negative Photoresist | Nippon Kayaku Co., Ltd. | SU-8 3050 | NA |
| Commercial assay or kit | Polydimethylsiloxane sylgard 184 | Dow Corning Corp. | SYLGARD 184 Silicone Elastomer Kit | NA |
| Commercial assay or kit | Barcoded Bead SeqB | ChemGenes Corporation | MACOSKO-2011–10 | NA |
| Commercial assay or kit | Maxima H Minus Reverse Transcriptase | Thermo Fisher Scientific | EP0751 | NA |

*Continued on next page*

*Continued*

| Reagent type (species) or resource | Designation | Source or reference | Identifiers | Additional information |
|---|---|---|---|---|
| Commercial assay or kit | Exonuclease I | New England Biolabs | M0293S | NA |
| Commercial assay or kit | KAPA HiFi HotStart ReadyMix | Roche Ltd. | KK2601 | NA |
| Commercial assay or kit | KAPA HiFi DNA polymerase | Roche Ltd. | KK2103 | NA |
| Commercial assay or kit | Agencourt AMPure XP beads | Beckman Coulter | A63882 | NA |
| Commercial assay or kit | DNA Clean and Concentrator Kit | Zymo Research | D4013 | NA |
| Commercial assay or kit | Qubit dsDNA HS Assay Kit | Thermo Fisher Scientific | Q32851 | NA |
| Commercial assay or kit | High-Capacity cDNA Reverse Transcription Kit | Thermo Fisher Scientific | 4368814 | NA |
| Commercial assay or kit | KOD SYBR qPCR | TOYOBO Co. Ltd. | QKD-201 | NA |
| Cell staining solution | May-Grünwald's eosin methylene blue solution modified | Merck KGaA | 101424 | NA |
| Cell staining solution | Giemsa's Azure Eosin Methylene Blue solution | Merck KGaA | 109204 | NA |
| Software, algorithm | Guppy v3.6.1 | Oxford Nanopore Technologies | NA | https://community.nanoporetech.com/ |
| Software, algorithm | MinKNOW v3.6.5 | Oxford Nanopore Technologies | NA | https://community.nanoporetech.com/ |
| Software, algorithm | TALC v1.01 | *Broseus et al., 2020a, Broseus et al., 2020b* | DOI: 10.1093/bioinformatics/btaa634 | https://gitlab.igh.cnrs.fr/lbroseus/TALC |
| Software, algorithm | rnaSPAdes v3.14.1 | *Bushmanova et al., 2019a, Bushmanova et al., 2019b* | DOI: 10.1093/gigascience/giz100 | https://cab.spbu.ru/software/rnaspades/ |
| Software, algorithm | Trinity 2.10.0 | *Grabherr et al., 2011, Grabherr et al., 2018* | DOI: 10.1038/nbt.1883 | https://github.com/trinityrnaseq/trinityrnaseq/wiki |
| Software, algorithm | EvidentialGene v2022.01.20 | *Gilbert, 2019* | NA | http://arthropods.eugenes.org/EvidentialGene/ |
| Software, algorithm | BUSCO v5.0.0 | *Seppey et al., 1962* | DOI: 10.1007/978-1-4939-9173-0_14 | https://busco.ezlab.org/ |
| Software, algorithm | Blast+ v2.2.31 | *Altschul et al., 1990; Camacho et al., 2009* | DOI: 10.1016/s0022-2836(05)80360-2 DOI: 10.1186/1471-2105-10-421 | https://ftp.ncbi.nlm.nih.gov/blast/executables/blast+/LATEST/ |
| Software, algorithm | Drop-seq tools v2.3.0 | McCarroll Lab *Wysoker et al., 2020* | NA | https://github.com/broadinstitute/Drop-seq |
| Software, algorithm | Picard Toolkit | Broad Institute *Picard Toolkit, 2019* | NA | http://broadinstitute.github.io/picard/ |
| Software, algorithm | STAR v2.7.8a | *Dobin et al., 2013; Dobin et al., 2021* | DOI: 10.1093/bioinformatics/bts635 | https://github.com/alexdobin/STAR |
| Software, algorithm | Surat v4.0.1 | *Butler et al., 2018; Stuart et al., 2019* | DOI: 10.1038/nbt.4096 DOI: 10.1016/j.cell.2019.05.031 | https://satijalab.org/seurat/ |
| Software, algorithm | Monocle 3 v0.2.3.0 | *Trapnell et al., 2014, Trapnell et al., 2021* | DOI: 10.1038/nbt.2859 | https://github.com/cole-trapnell-lab/monocle3 |

## Shrimp and cell preparation

Twenty-three female kuruma shrimp, *M. japonicus*, with an average weight of 20 g, were purchased from a local distributor and maintained in artificial seawater with a 34 ppt salinity with a recirculating

system at 20°C. Hemolymph was collected using an anticoagulant solution suitable for penaeid shrimp (*Söderhäll and Smith, 1983*) from an abdominal site. The collected hemolymph was centrifuged at $800 \times g$ for 10 min to collect the hemocytes, which were then washed twice with PBS, and the osmolarity was adjusted to kuruma shrimp (KPBS: 480 mM NaCl, 2.7 mM KCl, 8.1 mM $Na_2HPO_4 \cdot 12H_2O$, 1.47 mM $KH_2PO_4$, pH 7.4).

## Preparation of expressing gene list of hemocytes

De novo assembled transcript data were prepared as a reference for mapping Drop-seq data because the genome sequence of *M. japonicus* is still unknown. To improve the quality of the de novo assembled transcript sequences, we prepared long-read mRNA sequences using MinION (Oxford Nanopore Technologies) direct RNA sequencing to conduct hybrid de novo assembly. Poly (A) tailed RNA was purified from 58 µg of total RNA from the hemocytes of 16 shrimp using Dynabeads Oligo(dT)$_{25}$ (Thermo Fisher Scientific), and 500 ng of poly-(A) RNA was ligated to adaptors using a direct RNA sequencing kit (Oxford Nanopore Technologies) according to the manufacturer's manual version DRS_9080_v2_revL_14Aug2019. Finally, 44 ng of the library was obtained and sequenced using MinION by using a MinION flow cell R9.4.1 (Oxford Nanopore Technologies). All sequencing experiments were performed using MinKNOW v3.6.5 without base calling. Raw sequence data were then base-called using Guppy v3.6.1. Once the raw signal from the MinION fast5 files was converted into fastq files, the sequencing errors were corrected using TALC v1.01 (*Broseus et al., 2020a*, https://gitlab.igh.cnrs.fr/lbroseus/TALC, *Broseus et al., 2020b*), by using the Illumina short reads sequence of *M. japonicus* shrimp hemocytes (DDBJ Sequence Read Archive [DRA] accession number DRA004781). The corrected long-read sequences from MinION and short-read sequences from Illumina Miseq were hybrid de novo assembled using rnaSPAdes v3.14.1 (*Bushmanova et al., 2019a*, https://cab.spbu.ru/software/rnaspades/, *Bushmanova et al., 2019b*) and Trinity 2.10.0 (*Grabherr et al., 2011*, https://github.com/trinityrnaseq/trinityrnaseq/wiki, *Grabherr et al., 2018*). All assembled de novo transcripts were merged and subjected to the EvidentialGene program v2022.01.20 (http://arthropods.eugenes.org/EvidentialGene/, *Gilbert, 2019*) to remove similar sequences with a default parameter. The remaining sequences were renamed as Mj-XXX and were subjected to BUSCO analysis to check the quality of assembling (*Seppey et al., 1962*). The assembled transcripts were used as a hemocyte-expressing gene list. The assembled sequences and code used to perform base-calling and de novo assembly are available on GitHub at https://github.com/KeiichiroKOIWAI/Drop-seq_on_shrimp (*Koiwai, 2021a*, copy archived at swh:1:rev:cddd130b4cb841168bf3e29d0fbfef0de5ef2ad7, *Koiwai, 2021b*).

## Single-cell and single-bead encapsulation by a microfluidic device and exonuclease and reverse transcribe reaction on a bead

The Drop-seq procedure was used to encapsulate single hemocytes and single mRNA capture beads together into fL-scale microdroplets, as previously described (*Macosko et al., 2015*). The following steps were performed in triplicates for three shrimp individuals. Briefly, the self-built Drop-seq microfluidic device was prepared by molding polydimethylsiloxane (PDMS; Sylgard 184, Dow Corning Corp.) from the microchannel structure formed by the negative photoresist (SU-8 3050, Nippon Kayaku Co.). Using this device, droplets containing a cell and a Barcoded Bead SeqB (ChemGenes Corporation) were produced up to 2 mL per sample using a pressure pump system (Droplet generator, On-chip Biotechnologies Co., Ltd.). During the sample introduction, the vial bottles containing cells and beads were shaken using a vortex mixer to prevent sedimentation and aggregation (*Biočanin et al., 2019*). Droplets were collected from the channel outlet into the 50 mL corning tube and incubated at 80°C for 10 min in a water bath to promote hybridization of the poly(A) tail of mRNA and oligo d(T) on beads. After incubation, droplets were broken promptly and barcoded beads with captured transcriptomes were reverse transcribed using Maxima H Minus Reverse Transcriptase (Thermo Fisher Scientific) at room temperature (RT) for 30 min, then at 42°C for 90 min. Then, the beads were treated with Exonuclease I (New England Biolabs) to obtain single-cell transcriptomes attached to microparticles (STAMP). The first-strand cDNAs on beads were amplified using PCR. The beads obtained above were distributed throughout PCR tubes (1000 beads per tube), wherein 1× KAPA HiFi HS Ready Mix (KAPA Biosystems) and 0.8 µM 1st PCR primer were included in a 25 µL reaction volume. PCR amplification was achieved using the following program:

initial denaturation at 95°C for 3 min; 4 cycles at 98°C for 20 s, 65°C for 45 s, and 72°C for 6 min; 12 cycles of 98°C for 20 s, 67°C for 20 s, and 72°C for 6 min; and a final extension at 72°C for 5 min. The amplicons were pooled, double-purified with 0.9× AMPure XP beads (Beckman Coulter), and eluted in 100 µL of ddH₂O. Sequence-ready libraries were prepared according to *Picelli et al., 2014*. A total of 1 ng of each cDNA library was fragmented using home-made Tn5 transposome in a solution containing 10 mM TAPS-NaOH (pH 8.5), 5 mM MgCl₂, and 10% dimethylformamide at 55°C for 10 min. The cDNA fragments were purified using a DNA Clean and Concentrator Kit (Zymo Research) and eluted in 25 µL of ddH₂O. The index PCR reaction was performed by adding 12 µL of the elute to a mixture consisting of 1× Fidelity Buffer, 0.3 mM dNTPs, 0.5 U KAPA HiFi DNA polymerase (KAPA Biosystems), 0.2 µM P5 universal primer, and 0.2 µM i7 index primer. Each reaction was achieved as follows: initial extension and subsequent denaturation at 72°C for 3 min and 98°C for 30 s; 12 cycles of 98°C for 10 s, 63°C for 30 s, and 72°C for 30 s; and a final extension at 72°C for 5 min. The amplified library was purified using 0.9× AMPure XP beads and sequenced (paired-end) on an Illumina NextSeq 500 sequencer (NextSeq 500/550 High Output v2 kit [75 cycles]); 20 cycles for read1 with custom sequence primer, 8 cycles for index read, and 64 cycles for read2. Before performing the Drop-seq on *M. japonicus* shrimp hemocytes, we validated the protocol by performing the same procedure using a mixture of HEK293 and NIH3T3 cells and sequencing the test library using a Miseq Reagent Kit v3 (150 cycles).

## Analysis of single-cell data

Paired-end reads were processed and mapped to the reference de novo assembled gene list of hemocytes following the Drop-seq Core Computational Protocol version 2.0.0 and the corresponding Drop-seq tools v2.3.0 (https://github.com/broadinstitute/Drop-seq; *Wysoker et al., 2020*), provided by McCarroll Lab (http://mccarrolllab.org/dropseq/). The Picard suite (https://github.com/broadinstitute/picard, *Picard Toolkit, 2019*). was used to generate the unaligned bam files. The steps included the detection of barcode and UMI sequences, filtration and trimming of low-quality bases and adaptors or poly(A) tails, and the alignment of reads using STAR v2.7.8a (*Dobin et al., 2013*). The cumulative distribution of reads from the aligned bam files was obtained using BAMTagHistogram, and the number of cells was inferred using Drop-seq tools.

## Data integration

After digital expression data from three shrimps were read using Seurat v4.0.1 (*Butler et al., 2018*; *Hafemeister and Satija, 2019*) (https://satijalab.org/seurat/). To remove data from low-quality cells or empty droplets, we filtered digital expression data with unique feature counts over 4000 or less than 500 and mitochondrial counts over 10% or less than 1%, then the remained 2566 dataset was used for the further analysis. SCTransform (*Hafemeister and Satija, 2019*) was performed to remove the technical variation, while retaining biological heterogeneity. We ran a PCA using the expression matrix of the top 3000 most variable genes. The total number of principal components (PCs) required to compute and store was 50. The UMAP was then performed using the following parameters: n.neighbors, min.dist, and n.components were 50L, 0.2, and 2, respectively, to visualize the data in the two-dimensional space, and then the clusters were predicted with a resolution of 0.4.

## Functional prediction of the highly expressed genes and clusters

The de novo assembly constructed 30,986 transcripts. However, not all transcripts were not so expressed in our single-cell data. Thereby, we limited to the highly expressed genes as 'Ex50' that represent 50% of the total normalized expression data. The functions of Ex50 were predicted using BLAST program v2.2.31 (*Camacho et al., 2009*; *Altschul et al., 1990*) on three common penaeid shrimp, *L. vannamei*, *P. monodon*, and *M. japonicus*, proteins (totally 62,242 proteins were downloaded from NCBI Protein Groups on 10 April 2021) with the blastx parameter of e-value as 1e-6, and KOGs and GOs of each Ex50 were also predicted using TransDecoder (https://github.com/TransDecoder/TransDecoder, *Zimmermann et al., 2018*) and eggNOG-mapper (http://egg-nog-mapper.embl.de/, *Cantalapiedra et al., 2021*). From the annotated result, 21 KOGs and eleven GOs were selected, then average expressions were calculated on the clusters by AddModuleScore function of Seurat. The functions of each cluster were also predicted based on the marker genes.

Marker genes were predicted using the Seurat FindAllMarkers tool with the following parameters: min.pct as 0.5, logfc.threshold as 1, and test.

## Prediction of cell cycles based on *Drosophila* G2/M and S phase marker genes

To calculate cell cycle, Ex50 genes were annotated with *Drosophila* G2/M and S phase marker genes (https://github.com/hbc/tinyatlas, *Kirchner and Barrera, 2019*) using blastx parameter of e-value as 1e-6. Based on this result, we calculated cell cycle score of each single cell by Seurat CellCycleScoring function.

## Pseudo-temporal ordering of cells using Monocle 3

The integrated data of Seurat were transferred to Monocle 3 (*Trapnell et al., 2014*, https://github.com/cole-trapnell-lab/monocle3, *Trapnell et al., 2021*), to calculate a cell trajectory using the learn_graph function. We assigned the start point based on the expression of cell proliferation-related genes and *Drosophila* marker genes of the cell cycle.

## Comparison with *Drosophila* hemocyte marker genes

To check whether the *Drosophila* marker genes are applicable to shrimp, we performed a BLAST search on the *Drosophila* cell-type markers (https://github.com/hbc/tinyatlas, *Kirchner and Barrera, 2019*). Ex50 were blastx searched on *D. melanogaster* genes (dmel-all-gene-r6.34.fasta; downloaded from FlyBase https://flybase.org/) with the parameters of e-value as 1e-6.

## Visualization of cell growth-related and immune-related genes on single hemocytes

To visualize the immune-related genes of shrimp, we identified the distinct sequences from Ex50 based on their blastx results against penaeid shrimp proteins. Here, we focused on the cell growth-related genes and on immune-related genes, such as AMPs, lysozyme, clotting-related, melanization-related, phagocytosis-related, superoxide dismutase, Kazal-type proteinase inhibitors, and other immune-related genes listed in *Supplementary file 1*. The average expressions of each immune-related genes were calculated on the clusters by AddModuleScore function of Seurat.

## Cell sorting of hemocytes and qRT-PCR of marker genes

To validate the Drop-seq results on hemocytes, populations of hemocytes in the forward scatter (FSC) and side scatter (SSC) two-dimensional space were sorted using a microfluidic cell sorter (On-chip sort, On-chip Biotechnologies Co., Ltd.) from four shrimp individuals (*Figure 9—figure supplement 1*). In the FSC/SSC two-dimensional space, two main populations were predicted as R1: small/simple and R2: large/complexity populations, which were defined as agranulocytes and granulocytes, respectively. After sorting, some sorted hemocytes were immediately fixed in 2% formalin in KPBS and stained with a May-Grunwald and Giemsa staining solution to observe the cellular components. Briefly, fixed hemocytes were smeared on a glass slides and dried, stained for 5 min with 20% May-Grunwald stain solution (Merck KGaA) in 6.67 mM phosphate buffer (pH 6.6), washed with phosphate buffer, stained for 15 min with 4% Giemsa stain solution (Merck KGaA) in 6.67 mM phosphate buffer (pH 6.6), and washed with tap water, dried, mounted with malinol (Muto Pure Chemicals). Non-stained and stained hemocytes were subjected to microscopy IX71 (Olympus Corporation) to observe their structures.

Total RNA was also collected from sorted cells and pre-sorted cells. The concentration of RNA was measured using a nanodrop, and cDNA was transcribed using a High-Capacity cDNA Reverse Transcription Kit (Thermo Fisher Scientific). Constructed cDNA was diluted five times with TE buffer and subjected to qRT-PCR with a number of cycles as 40 using KOD SYBR qPCR (TOYOBO Co. Ltd.), following the manufacturer's protocol. When the CT value was not detectable, we set a CT value of 40 to use for $\Delta$CT calculation. The expression of each gene was calculated using the $\Delta\Delta$CT method against elongation factor-1 alpha and total hemocytes.

## Statistics

Statistics of qRT-PCR were performed on R software. Significance between two hemocyte R1 and R2 was calculated by unpaired t-test. The p values shown in the figures are represented by *p<0.05.

## Data files and analysis code

The raw sequence data of newly sequenced *M. japonicus* transcriptomic reads were archived in the DDBJ Sequence Read Archive (DRA) of the DNA Data Bank of Japan as follows: MinION mRNA direct sequencing: DRA010948; Drop-seq shrimp rep1: DRA010950; shrimp rep2: DRA010951; shrimp rep3: DRA010952; mixture sample of HEK293 and 3T3 cells: DRA010949. The Seurat digital expression data were archived in the Genomic Expression Archive of the DNA Data Bank of Japan: E-GEAD-403. Fast5 data of MinION direct RNA sequencing will be made available upon request from the authors. The code used to perform de novo assembly, clustering, and marker analysis is available on GitHub at https://github.com/KeiichiroKOIWAI/Drop-seq_on_shrimp, *Koiwai, 2021a*.

## Acknowledgements

We would like to thank Fumiko Sunaga for her technical support in performing cell sorting and analysis, and Editage (http://www.editage.com) for English-language editing. This work was supported by JSPS KAKENHI Grant-in-Aid for Early-Career Scientists (20K15603) and Grant-in-Aid for JSPS Fellows (19J00539) to K Koiwai; Grant-in-Aid for Scientific Research on Innovative Areas (17H06425) to K Kikuchi.

## Additional information

### Competing interests

Soichiro Tsuda: is an employee of bitBiome inc, and does not own any stock of the company. The other authors declare that no competing interests exist.

### Funding

| Funder | Grant reference number | Author |
| --- | --- | --- |
| Japan Society for the Promotion of Science | 20K15603 | Keiichiro Koiwai |
| Japan Society for the Promotion of Science | 19J00539 | Keiichiro Koiwai |
| Japan Society for the Promotion of Science | 17H06425 | Kiyoshi Kikuchi |

The funders had no role in study design, data collection and interpretation, or the decision to submit the work for publication.

### Author contributions

Keiichiro Koiwai, Conceptualization, Resources, Data curation, Formal analysis, Funding acquisition, Investigation, Visualization, Writing - original draft, Project administration, Writing - review and editing; Takashi Koyama, Resources, Investigation, Writing - review and editing; Soichiro Tsuda, Investigation, Writing - review and editing; Atsushi Toyoda, Resources, Investigation; Kiyoshi Kikuchi, Resources, Supervision, Funding acquisition; Hiroaki Suzuki, Ryuji Kawano, Resources, Supervision, Writing - review and editing

### Author ORCIDs

Keiichiro Koiwai  https://orcid.org/0000-0002-8890-8229
Takashi Koyama  https://orcid.org/0000-0001-7140-7244
Atsushi Toyoda  http://orcid.org/0000-0002-0728-7548

Hiroaki Suzuki [ID] https://orcid.org/0000-0002-8899-0955
Ryuji Kawano [ID] https://orcid.org/0000-0001-6523-0649

### Decision letter and Author response
Decision letter https://doi.org/10.7554/eLife.66954.sa1
Author response https://doi.org/10.7554/eLife.66954.sa2

## Additional files

### Supplementary files
• Supplementary file 1. Table representing blastx researching of assembled genes against penaeid shrimp's proteins.

• Supplementary file 2. Table representing predicted marker genes per cluster.

• Supplementary file 3. Table representing blastx researching of assembled genes against *Drosophila* cell cycle markers.

• Supplementary file 4. Table representing blastx researching of assembled genes against *Drosophila* cell-type markers.

• Transparent reporting form

### Data availability
Sequencing data have been deposited in DDBJ under accession codes DRA010948, DRA010949, DRA010950, DRA010951, and DRA010952. Digital expression data of Drop-seq from three shrimp were archived in DDBJ under accession code E-GEAD-403. Data code can be accessed at https://github.com/KeiichiroKOIWAI/Drop-seq_on_shrimp (copy archived at https://archive.softwareheritage.org/swh:1:rev:cddd130b4cb841168bf3e29d0fbfef0de5ef2ad7). All data generated or analyzed during this study are included in the manuscript and supporting files. All tables are provided as Supplementary files. Source data files are provided to support the sinaplots in Figure 2, Figure 2-figure supplement 1. Source data files are provided to support thepercentage of cell state in Figure 5. Source data file is provided to support the bar graph in Figure 9F.

The following datasets were generated:

| Author(s) | Year | Dataset title | Dataset URL | Database and Identifier |
|---|---|---|---|---|
| Koiwai K, Koyama T, Tsuda S, Toyoda A, Kikuchi K, Suzuki H, Kawano R | 2020 | Direct RNA sequencing data of hemocytes of Marsupenaeus japonicus using MinION | https://trace.ddbj.nig.ac.jp/DRASearch/submission?acc=DRA010948 | DDBJ Sequence Read Archive, DRA010948 |
| Koiwai K, Koyama T, Tsuda S, Toyoda A, Kikuchi K, Suzuki H, Kawano R | 2020 | Raw sequence data of drop-seq on shrimp 1 | https://trace.ddbj.nig.ac.jp/DRASearch/submission?acc=DRA010950 | DDBJ Sequence Read Archive, DRA010 950 |
| Koiwai K, Koyama T, Tsuda S, Toyoda A, Kikuchi K, Suzuki H, Kawano R | 2020 | Raw sequence data of drop-seq on shrimp 2 | https://trace.ddbj.nig.ac.jp/DRASearch/submission?acc=DRA010951 | DDBJ Sequence Read Archive, DRA010 951 |
| Koiwai K, Koyama T, Tsuda S, Toyoda A, Kikuchi K, Suzuki H, Kawano R | 2020 | Raw sequence data of drop-seq on shrimp 3 | https://trace.ddbj.nig.ac.jp/DRASearch/submission?acc=DRA010952 | DDBJ Sequence Read Archive, DRA010 952 |
| Koiwai K, Koyama T, Tsuda S, Toyoda A, Kikuchi K, Suzuki H, Kawano R | 2020 | Digital expression data of Drop-seq from three shrimp | ftp://ftp.ddbj.nig.ac.jp/ddbj_database/gea/experiment/E-GEAD-000/E-GEAD-403 | DDBJ Genomic Expression Archive, E-GEAD-403 |
| Koiwai K, Koyama T, Tsuda S, Toyoda A, Kikuchi K, Suzuki H, Kawano R | 2020 | Raw sequence data of drop-seq on mixture sample of HEK293 and 3T3 cells | https://trace.ddbj.nig.ac.jp/DRASearch/submission?acc=DRA010949 | DDBJ Sequence Read Archive, DRA010949 |

The following previously published dataset was used:

| Author(s) | Year | Dataset title | Dataset URL | Database and Identifier |
|---|---|---|---|---|
| Koiwai K, Alenton RRR, Shiomi R, Nozaki R, Kondo H, Hirono I | 2016 | Transcriptome analysis on different sub-populations of hemocytes of kuruma shrimp Marsupenaeus japonicus | https://trace.ddbj.nig.ac.jp/DRASearch/submission?acc=DRA004781 | DDBJ Sequence Read Archive, DRA004781 |

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
