## [Decision Letter]

**Acceptance summary:**

Unlike in classical model organisms, it has been challenging to comprehensively understand the biological attributes of cells in non-model systems such as in shrimps. This study takes advantage of single-cell RNA sequencing and profiles the diversity and putative lineages of shrimp hemocytes. The primary claims are well supported by the data, and this study will contribute substantially, not only to crustacean biology but also related areas.

**Decision letter after peer review:**

Thank you for submitting your article "Single-cell RNA-seq analysis reveals penaeid shrimp hemocyte subpopulations and cell differentiation process" for consideration by *eLife*. Your article has been reviewed by 3 peer reviewers, one of whom is a member of our Board of Reviewing Editors, and the evaluation has been overseen by Utpal Banerjee as the Senior Editor. The reviewers have opted to remain anonymous.

Essential Revisions:

The reviewers agreed that the study was well-performed and analyzed and bears the potentials to improve understandings of shrimp immune systems. Reviewers raised several concerns and recognized that additional data QC is essential to make strong conclusions. Requirements on in vivo validation will further strengthen the paper; however, the reviewers agreed it would not be feasible to perform the experiments suggested. Please find the list below and details in reviewers' comments.

Data QC:

1. de novo assembly and correlation to reference genome

2. Cell viability and low UMI

3. Batch effects (similarities between triplicates) and doublets

4. Cutoff values for the number of UMI per cell

5. Mitochondrial contents

6. Additional analysis for subcluster clustering including Hem1 and Hem6

7. Re-validation of pseudotime and lineage analyses

in vivo validation:

The reviewers acknowledged that it is difficult to do in-depth in vivo verification with shrimps. The working model (e.g. Figure 7) could be modified or removed if additional in vivo validations are not provided. If possible, in vivo experiments could be supplied to keep the model.

*Reviewer #1 (Recommendations for the authors):*

1. Page 5 line 88, Cho et al. Nature Communications (2020) [PMID 32900993] sequenced *Drosophila* hemocytes in various conditions using the Drop-Seq platform.

2. Page 11 line 162, typo; TINGAL to TINAGL

3. In Figure 3B, please consider adding pseudotime density plots for each Hem cluster. I think this can clearly show where Hem clusters are located in the trajectory.

4. In Figure 6F, the authors showed that a subset of marker genes are differentially expressed in R1 or R2 groups with the use of qRT-PCR. However, it is not clear the data is statistically sound to claim the result. Additional statistical analysis is required to adequately support the authors' claim.

5. The authors used "top 3,000 most variable genes" in the PCA while the data have 3,334 commonly expressed genes. Are all these variable genes included in the "common genes"? This point should be confirmed.

*Reviewer #2 (Recommendations for the authors):*

It would help a reader to analyze the data if the *Drosophila* genes corresponding to the Mj genes in figure3 supplement 1 and figure 3 supplement 2 were included at the side in the "table".

Figure 7 is too speculative since no functional experiments (as RNAi) has been done to confirm the schedule, and therefore figure 7 should be removed.

The quality check should also be discussed more as well as the low number of UMI, and the percentage of mitochondrial genes should be included.

The last part with FACS sorting and qPCR could be removed, since this doesn't add any information or confirmation. The proposed lineages could also be discussed in a bit more critical way.

*Reviewer #3 (Recommendations for the authors):*

The following suggestions might help the authors to strength the science or improve the manuscript:

1. There is a small cluster of cells of Hem1 up of the label of 'Hem1' in Figure 1, but this small cluster was gone in all other analysis. Can you discuss about this particular cluster?

2. Previous study indicates phagocytosis by hemocyte is a crucial defense mechanism for the host against infections (PMID: 32194551). Those phagocytotic hemocytes should be terminally differentiated cells, and phagocytosis related genes should express in those cells. Can you check how those genes are expressed in the shrimp hemocytes? This could further confirm your differentiation hypothesis.

3. Two additional *Drosophila* hemocyte single-cell RNA sequencing papers should be cited (PMID: 32487456; PMID: 32162708).

4. The method used for the dye staining in Figure 6B and D should be presented in the Materials and methods part.

5. The primer sequence used for qRT-PCR should be presented in the Materials and methods part.

[Editors' note: further revisions were suggested prior to acceptance, as described below.]

Thank you for resubmitting your work entitled "Single-cell RNA-seq analysis reveals penaeid shrimp hemocyte subpopulations and cell differentiation process" for further consideration by *eLife*. Your revised article has been evaluated by Utpal Banerjee (Senior Editor) and a Reviewing Editor.

The manuscript has been improved but there are some remaining issues that need to be addressed, as outlined below:

As you can find below comments, the reviewers agreed that the authors addressed most of the concerns by reanalyzing and processing the data. Yet, the new data raise additional points with regards to (1) considerably different conclusions obtained by alternative parameters and (2) multiple calls of the same gene transcripts in some analyses due to the lack of reference genome. These points need to be reanalyzed if possible or at least be discussed.

*Reviewer #1 (Recommendations for the authors):*

The authors performed additional analyses and successfully addressed all the major concerns. The authors substantially enhanced the quality of data by removing dead cells and increasing UMI/gene counts. Although overall numbers of UMI and genes are still lower than those of other model organisms, it would be potentially improved with a reference genome which is expected to be built shortly. The quality of clustering is also clear enough. Further, descriptions of in vivo experiments and models are simplified as suggested, and I hope to see in-depth validation in the near future.

*Reviewer #2 (Recommendations for the authors):*

The authors have responded to all questions but my main concern is that when they remade the analysis nearly completely, the results were so different with a new software and when new parameters are used. This is a bit worrying especially for the lineage determination. In the first version the conclusion was four lineages and now only two lineages. The authors need to explain why the resolution is so very different.

A problem that should at least be discussed is related to the lack of genome sequence, and this is evident for example in figure 8 figure supplement 1, where several transcripts seem to be from the same gene (AMH87234.1 = Mj-18245 + Mj-19281+ Mj-20968 and ABW88999.1=Mj-3787+Mj-19067+Mj-19338+Mj-28125). This is always a problem when using transcriptome instead of a fully annotated genome, and it is important to be aware of that single-cell RNA-seq provides only a snapshot of each individual cell and they may be in different stages of the cell cycle and / or turnover of RNA. Therefore, transcripts belonging to the same gene should be merged when possible. I realize that this is impossible for all transcripts, but for known important immune genes and/or cell cycle genes used in the clustering and lineage determination it should be done.

Regarding the validation at line 492 and forward, this is still not meaningful. The authors say that BrdU incorporation and in situ hybridization are not possible for shrimp hemocytes, but there are several published studies showing in situ hybridization of crustacean hemocytes so this answer is difficult to understand. The FACS-analysis is not a good enough separation, and in situ hybridization of some of the transcripts could be of value to show the morphology of the corresponding cell. Maybe morphology doesn't tell us so much, but this is important to show.

*Reviewer #3 (Recommendations for the authors):*

The authors did a great job in the revised manuscript to address all the concerns that I had for the first submission.

---

## [Author Response]

Essential Revisions:The reviewers agreed that the study was well-performed and analyzed and bears the potentials to improve understandings of shrimp immune systems. Reviewers raised several concerns and recognized that additional data QC is essential to make strong conclusions. Requirements on in vivo validation will further strengthen the paper; however, the reviewers agreed it would not be feasible to perform the experiments suggested. Please find the list below and details in reviewers' comments.

Thank you for the review comments and the summary. We have also examined the percentage of mitochondria-derived sequences and verified why there were so few UMIs in one cell. A detailed response is given below. We believe that this correction answers your point.

Data QC:1. de novo assembly and correlation to reference genome

The genome sequence of the kuruma shrimp *M. japonicus* has only been registered, and the high-quality data has not been published yet. Therefore, we could not perform validation using the genome sequence. However, by applying the BUSCO tool to the assembled sequences, we verified the quality of the assembly genes.

2. Cell viability and low UMI

By confirming the mitochondrial-derived sequences, we cleared up the suspicion that large numbers of dead cells were contaminating. We have also succeeded in increasing the number of UMIs by changing mapping software and adjusting the parameters. The value of UMIs is still lower than that of other model organisms, but we think that will improve as the reference genome is published in the future. I have discussed this in the manuscript.

3. Batch effects (similarities between triplicates) and doublets

The triplicate distribution after batch correction is shown in the Figure 2—figure supplement 1 and Figure 5—figure supplement 1. For Doublets, we assumed UMI less than 4000 this time because none of them had prominent UMI.

4. Cutoff values for the number of UMI per cell

Since there are no set rules here, we set the parameters for one cell (in the same way as in comment 3) based on the initial distribution diagram.

5. Mitochondrial contents

As we answered in the *cell viability and low UMI* section, we checked and figured the results of mitochondrial contents.

6. Additional analysis for subcluster clustering including Hem1 and Hem67. Re-validation of pseudotime and lineage analyses

Based on the new UMI counts, we re-did *in silico* Clustering and pseudotime analysis with new parameters.

in vivo validation:The reviewers acknowledged that it is difficult to do in-depth in vivo verification with shrimps. The working model (e.g. Figure 7) could be modified or removed if additional in vivo validations are not provided. If possible, in vivo experiments could be supplied to keep the model.

We strongly agree this comment, it is necessary to check how hemocytes are differentiated and classified by using marker antibodies, in situ, etc., but it is difficult to perform that experiment accurately in shrimp immediately. As commented by the editor, the figure here has been removed.

Reviewer #1 (Recommendations for the authors):1. Page 5 line 88, Cho et al. Nature Communications (2020) [PMID 32900993] sequenced *Drosophila* hemocytes in various conditions using the Drop-Seq platform.

Thank you for your comment. We have added it to the reference.

2. Page 11 line 162, typo; TINGAL to TINAGL

Thank you, we removed the sentence.

3. In Figure 3B, please consider adding pseudotime density plots for each Hem cluster. I think this can clearly show where Hem clusters are located in the trajectory.

Thank you for useful comments. We have made new figures for pseudotime analysis. (Figure 5A and B).

4. In Figure 6F, the authors showed that a subset of marker genes are differentially expressed in R1 or R2 groups with the use of qRT-PCR. However, it is not clear the data is statistically sound to claim the result. Additional statistical analysis is required to adequately support the authors' claim.

We did t-test for all qRT-PCR result and put results on the figure.

5. The authors used "top 3,000 most variable genes" in the PCA while the data have 3,334 commonly expressed genes. Are all these variable genes included in the "common genes"? This point should be confirmed.

We have re-done the previous analysis and changed to analyze the genes that account for 50% of the expression level, as previous analysis we still used "top 3,000 most variable genes" in the PCA.

Reviewer #2 (Recommendations for the authors):It would help a reader to analyze the data if the *Drosophila* genes corresponding to the Mj genes in figure3 supplement 1 and figure 3 supplement 2 were included at the side in the "table".

Thank you for suggestion. We figured the relationship between Mj genes and *Drosophila* genes corresponding as Figure 6.

Figure 7 is too speculative since no functional experiments (as RNAi) has been done to confirm the schedule, and therefore figure 7 should be removed.

We understood your comment. Since we could not conduct any functional research about marker genes, we have removed figure 7.

The quality check should also be discussed more as well as the low number of UMI, and the percentage of mitochondrial genes should be included.

The number of UMIs/cell has been improved as a result of changes to the mapping software and the parameters used when analyzing the UMI detection method for each cell. This is still lower than human cells, but we believe it is the best we can do now that genomic information is not available. The ratio of mitochondria is calculated and summarized in the Figure 2.

The last part with FACS sorting and qPCR could be removed, since this doesn't add any information or confirmation. The proposed lineages could also be discussed in a bit more critical way.

What we want to show here is that it is very difficult to classify hemocytes by morphologically, and even if we could, it is likely to be divided into two rough groups (FACS result). As in the answer to the question above, we believe the advantage of this project is that we were able to search for marker candidates and provide guidelines for cell classification in the future. Of course, in the future, we hope to look at the function and expression of each gene.

Reviewer #3 (Recommendations for the authors):The following suggestions might help the authors to strength the science or improve the manuscript:1. There is a small cluster of cells of Hem1 up of the label of 'Hem1' in Figure 1, but this small cluster was gone in all other analysis. Can you discuss about this particular cluster?

Based on the new UMI counts, we re-did *in silico* Clustering and pseudotime analysis with new parameters. It made more clear result.

2. Previous study indicates phagocytosis by hemocyte is a crucial defense mechanism for the host against infections (PMID: 32194551). Those phagocytotic hemocytes should be terminally differentiated cells, and phagocytosis related genes should express in those cells. Can you check how those genes are expressed in the shrimp hemocytes? This could further confirm your differentiation hypothesis.

Thank you for your constructive feedback. We discussed about phagocytosi-related genes in this study.

3. Two additional *Drosophila* hemocyte single-cell RNA sequencing papers should be cited (PMID: 32487456; PMID: 32162708).

Thank you for the input. We added the reference on revised manuscript.

4. The method used for the dye staining in Figure 6B and D should be presented in the Materials and methods part.

Thank you for the suggestion. We added the protocol.

5. The primer sequence used for qRT-PCR should be presented in the Materials and methods part.

Yes, thank you. The sequences are on the Key Resource Table.

[Editors' note: further revisions were suggested prior to acceptance, as described below.]The manuscript has been improved but there are some remaining issues that need to be addressed, as outlined below:As you can find below comments, the reviewers agreed that the authors addressed most of the concerns by reanalyzing and processing the data. Yet, the new data raise additional points with regards to (1) considerably different conclusions obtained by alternative parameters and (2) multiple calls of the same gene transcripts in some analyses due to the lack of reference genome. These points need to be reanalyzed if possible or at least be discussed.

Thank you for the review comments and the summary. As for the comment (1), we believe that this is due to the fact that the number of UMIs detections has increased, allowing for a more accurate analysis. As for the comment (2), this is true, but we will not know which genes are important until we map and analyze them. If we summarized only specific genes by ourselves, it would be a bias, so we conducted the analysis based on this result. A detailed response is given below. Thanks again to your and reviewers’ comments, we were able to make our paper better. We believe that this revision will meet your expectations.

Reviewer #1 (Recommendations for the authors):The authors performed additional analyses and successfully addressed all the major concerns. The authors substantially enhanced the quality of data by removing dead cells and increasing UMI/gene counts. Although overall numbers of UMI and genes are still lower than those of other model organisms, it would be potentially improved with a reference genome which is expected to be built shortly. The quality of clustering is also clear enough. Further, descriptions of in vivo experiments and models are simplified as suggested, and I hope to see in-depth validation in the near future.

Thank you for carefully reading our revised manuscript. We think your comments helped us to improve our manuscript. We will continue mapping to the genome sequence in near future and conduct more detailed single-cell analysis based on this manuscript.

Reviewer #2 (Recommendations for the authors):The authors have responded to all questions but my main concern is that when they remade the analysis nearly completely, the results were so different with a new software and when new parameters are used. This is a bit worrying especially for the lineage determination. In the first version the conclusion was four lineages and now only two lineages. The authors need to explain why the resolution is so very different.

Thank you for deep reading our revised manuscript and your all comments improved our revised manuscript. This is probably due to the improvement in the number of detected UMIs. The early version showed four pathways because the resolution of the analysis was low, and we think the newer version, with its improved resolution of gene analysis per cell, is the best for the current situation. Also, the endpoints of the previous four pathways are thought to be enclosed in the current pathway. We do not believe that our result is complete, and we expect that more detailed studies will be conducted in the future as reference genomes become available.

A problem that should at least be discussed is related to the lack of genome sequence, and this is evident for example in figure 8 figure supplement 1, where several transcripts seem to be from the same gene (AMH87234.1 = Mj-18245 + Mj-19281+ Mj-20968 and ABW88999.1=Mj-3787+Mj-19067+Mj-19338+Mj-28125). This is always a problem when using transcriptome instead of a fully annotated genome, and it is important to be aware of that single-cell RNA-seq provides only a snapshot of each individual cell and they may be in different stages of the cell cycle and / or turnover of RNA. Therefore, transcripts belonging to the same gene should be merged when possible. I realize that this is impossible for all transcripts, but for known important immune genes and/or cell cycle genes used in the clustering and lineage determination it should be done.

We understand this concern you pointed. However, we will not know which genes are important until we map and analyze them. We also used a program called EvidentialGene to exclude similar genes as much as possible, and we tried to exclude duplicates using a program called CD-HIT, but it did not work as well as EvidentialGene. As you pointed out, if we summarized only specific genes by ourselves, it would be a bias, so we conducted the analysis based on the results of clustering by EvidentialGene in this manuscript. We have added sentences in the main text to help the reader understand our thoughts on this point.

Regarding the validation at line 492 and forward, this is still not meaningful. The authors say that BrdU incorporation and in situ hybridization are not possible for shrimp hemocytes, but there are several published studies showing in situ hybridization of crustacean hemocytes so this answer is difficult to understand. The FACS-analysis is not a good enough separation, and in situ hybridization of some of the transcripts could be of value to show the morphology of the corresponding cell. Maybe morphology doesn't tell us so much, but this is important to show.

Yes, we think this is an important point. Also, we strongly agree with the "Maybe morphology doesn't tell us so much" part. However, we believe that the significance of this paper lies in the fact that we applied single cell analysis to shrimp hemocytes and established guidelines for a classification. The actual function of the markers and differentiation factors discovered in this study is the target of future research, and we would like to challenge more molecular biological assay systems at that time. We believe that the reader will understand the challenges of this research result, as we have also described them in the summary section after Line 595.